# LLMs Boost the Performance of Decision Trees on Tabular Data across Sample Sizes

## Abstract

Large language models (LLMs) perform remarkably well on tabular datasets in zero- and few-shot settings, since they can extract meaning from natural language column headers that describe features and labels. In contrast to LLMs, gradient-boosted decision trees (GBDTs) must learn the relationships among columns from scratch, increasing their data requirements. Meanwhile, LLMs are not competitive with GBDTs on medium or large datasets, and their scalability is capped by their limited context lengths. In this paper, we propose LLM-Boost, a simple and lightweight approach for fusing large language models with gradient-boosted decision trees, which enables larger datasets to benefit from the natural language capabilities of LLMs than was previously shown. While matching LLMs at sufficiently small dataset sizes and GBDTs at sufficiently large sizes, LLM-Boost outperforms both standalone models on a wide range of dataset sizes in between. We demonstrate state-of-the-art performance against numerous baselines and ensembling approaches, and we also show how to fuse GBDTs with TabPFN, a recent non-LLM model for in-context learning on tabular data. We find that this combination achieves the best performance on larger datasets. We release our code at `https://anonymous.4open.science/r/LLM-Boost-21DD`.

## 1 Introduction

Tabular data, or spreadsheets, constitute a large portion of real-world machine learning problems (Borisov et al., 2022). Tabular data comprise (a) columns, each containing a different feature or label; (b) rows, each containing an individual data sample; and (c) column headers describing the content of each column, often in the form of text.

Gradient-boosted decision trees (GBDTs), such as XGBoost (Chen & Guestrin, 2016), LightGBM (Ke et al., 2017), and CatBoost (Prokhorenkova et al., 2018), have remained the de facto machine learning algorithms for analyzing tabular data over the past decade (McElfresh et al., 2024). They are efficient to train even on a CPU; they achieve competitive performance on a wide variety of datasets and sample sizes'. However, GBDTs have a major drawback: they only ingest the row features in a table and not the column headers, which may contain useful text descriptions. For example, one may not need training data to anticipate that a hospital patient's weight is useful for predicting occurrences of heart disease. Instead of leveraging column headers, from which a human might intuit relationships between columns, GBDTs have to learn these relationships from scratch from the feature values themselves.

In contrast to GBDTs, large language models (LLMs) can parse and extract meaning from column headers, enabling them to achieve superior performance to GBDTs on very small tabular datasets with interpretable headers (Hegselmann et al., 2023). LLMs can even make accurate zero-shot predictions solely by applying natural language understanding to column headers without any training samples at all (Hegselmann et al., 2023). Despite their ability to parse column headers, LLMs are severely limited by their limited context length and high fine-tuning costs. Moreover, LLMs make poor use of large sample sizes, whereas GBDTs scale well to massive datasets.

In this paper, we combine the strengths of gradient-boosted decision trees and large language models to build models that simultaneously possess natural language understanding and use column headers, additionally scaling to much larger tabular datasets than LLMs could alone. Our method, LLM-Boost, uses LLM predictions as a starting point for GBDT algorithms, and then learns the residuals

from LLM predictions to the label. This technique allows us to not only use the column headers for a strong prior but also benefits from the strong inductive bias and scalability of decision tree algorithms. In our experiments, LLM-Boost showcases state-of-the-art performance, outcompeting strong baselines including both single models and other ensemble approaches, across a large range of dataset sizes. LLM-Boost excels at small and medium sized datasets that are too large for LLMs yet not large enough that GBDTs do not benefit from column headers. For scenarios where interpretable column headers do not exist, we apply the same boosting approach except swapping out LLMs for TabPFN (Hollmann et al., 2023a), a recent in-context learning model that demonstrates strong performance on small tabular datasets but lacks scalability due to its limited context length. Our boosted TabPFN combination achieves the top performance among all methods we consider outside of the very small dataset regime where our boosted LLMs reign supreme.

We summarize our contributions as follows:

- We propose LLM-Boost: a novel yet simple and easy-to-implement boosting mechanism that combines LLMs, which ingest semantic column headers, with GBDTs that can scale to massive datasets.
- We conduct thorough experiments across numerous datasets and sample sizes, comparing to strong baselines. LLM-Boost demonstrates consistently strong performance.
- As additional studies, we show how to fuse TabPFN and GBDTs for performance gains over GBDTs alone across dataset sizes without using column headers.

## 2 RELATED WORK

### 2.1 GBDTs FOR TABULAR DATA

Gradient boosted decision tree algorithms such as XGBoost (Chen & Guestrin, 2016), Catboost (Prokhorenkova et al., 2018) and LightGBM (Ke et al., 2017) offer state-of-the-art or near state-of-the-art performance on many tabular tasks (Grinsztajn et al., 2022). Compared to deep learning models with similar performance, GBDTs offer faster training and inference speeds even without GPUs, are easy to tune, and are more straightforward to interpret. However, when compared to deep learning models, tree based models do not generalize as well to diverse unseen data and are not as robust to uninformative features (Grinsztajn et al., 2022). Recently, TabPFN (Hollmann et al., 2023a), a transformer for tabular in-context learning has demonstrated superior performance on small datasets (McElfresh et al., 2024). In our work, we adopt GBDTs as a base model due to their ability to benefit from large volumes of data, and we augment them with TabPFN and LLMs using boosting.

### 2.2 BOOSTING

Boosting is an ensembling technique for combining multiple weak learners to form a single strong prediction model (Freund & Schapire, 1997). Boosting algorithms are sequential processes whereby new learners are progressively added to predict the residual error of the current ensemble until the error becomes sufficiently small. Gradient boosting additionally provides a mechanism to update the new learners using an arbitrary differentiable loss function via gradient descent (Friedman, 2001). Although there are implementation differences in the GBDT algorithms mentioned above, they share the fundamental process of making predictions using an ensemble of weak decision tree models.

### 2.3 LLMs FOR TABULAR DATA

Large language models (LLMs) are trained on vast and diverse datasets, enabling them to solve a wide variety of problems, especially in zero- or few-shot settings (Hegselmann et al., 2023). Recent works have successfully repurposed LLMs for tabular data related tasks such as table understanding (Chen, 2023), tabular representation learning (Iida et al., 2021; Chen et al., 2023), time series forecasting (Gruver et al., 2023), and quantitative reasoning (Sui et al., 2023).

Repurposing LLMs for tabular prediction tasks requires data serialization and prompt engineering. Data serialization is required as LLMs are sequence to sequence models. While direct serialization of the values in a row is possible, converting rows into meaningful human-readable sentences

containing the row values and the column headers together aids the LLM in understanding the rows. Prompt engineering methods such as task descriptions and in-context examples as well as fine-tuning the LLM on the tabular prediction task itself can improve the model's domain-specific abilities.

Although approaches such as in-context examples and task specific fine-tuning enable the model to see more tabular examples, they come with drawbacks. LLMs are bottle-necked by context length limits, so it is difficult to provide more than a few in-context examples. Additionally, fine-tuning requires considerable computational overhead, even on simple tabular prediction tasks, and often underperforms alternatives such as GBDTs on larger datasets (Dinh et al., 2022) (Hegselmann et al., 2023).

Alternatively, LLMs have been used for automatic feature engineering in the tabular domain. Lightweight models, such as GBDTs, that are then trained on the augmented set of features have demonstrated superior performance to those trained on the original features Hollmann et al. (2023b); Nam et al. (2024). While this approach is computationally efficient at inference-time compared to our proposed procedure which uses the LLM during inference, the LLM typically only utilizes a small fraction of the table's samples to generate new features. Additionally, this approach usually requires powerful API models to be effective Hollmann et al. (2023b).

## 2.4 TABPFN

TabPFN (Hollmann et al., 2023a) is a transformer based network for tabular data, which is trained offline once to approximate Bayesian inference on synthetic datasets drawn from a prior. TabPFN performs in-context learning on the whole trainset, which does not require any parameter updates and can make predictions for the entire testset in a single forward pass. Superior speed and performance of TabPFN makes it ideal for datasets with up to 1000 samples. However, dataset size limitations remain a significant downside when adopting this method.

## 2.5 ENSEMBLING DIFFERENT MODEL CLASSES FOR TABULAR DATA

Due to the contrasting strengths and weaknesses of tree-based algorithms, traditional deep learning methods and LLMs for tabular data practitioners often use ensembles for more stable predictions. The predominant ensemble approach is feature stacking (Levin et al., 2023), where predictions of one model are used as features for the next. Efficient fusion of different model classes for tabular data is still an open problem.

## 3 METHOD

In this section, we detail the LLM-Boost algorithm, which is depicted in Figure 1. Broadly, LLM-Boost first takes a tabular dataset and extracts LLM scores, or logits, for each row of the table. We then augment a GBDT model by seeding it with the LLM's logits so that it learns the residual to the labels. When we perform this procedure, we must carefully tune a scaling parameter so that the GBDT is not overly reliant on these LLM predictions but simultaneously does not ignore them. Our approach is equivalent to replacing the first tree of the GBDT ensemble with the static prediction of the LLM which need to be pre-computed only once for inference and training. We then fit the GBDT to the residuals and evaluate the combined model's classification performance. We detail our pipeline in the following sections.

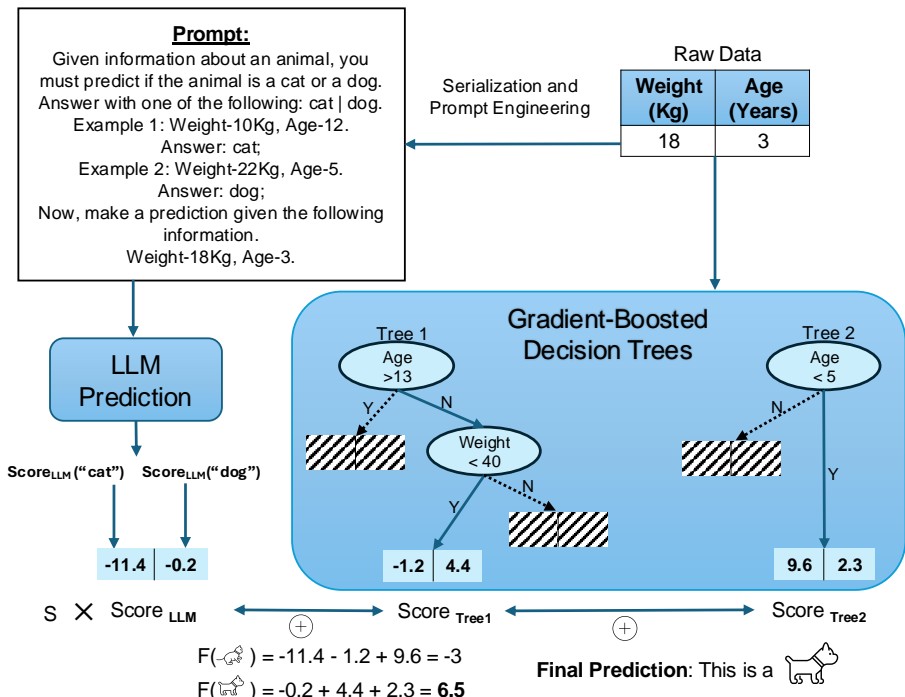

Figure 1: How **LLM-Boost works for a toy cat vs. dog classification problem.** Note that here the selected nodes are denoted in light blue. The scaling parameter denoted by S allows for controlling the effect of the LLM predictions on the tree ensemble.

## 3.1 EXTRACTING LLM SCORES

Our first step is to extract the LLM predictions for each row of a given tabular dataset. We create simple natural language, few-shot prompts utilizing the prompt generation and serialization tools developed by Slack & Singh (2023). The prompts are designed so that an instruction-tuned LLM will output one of the classification labels for each row of data. An example prompt for the UCI adult income dataset is given below.

> **Example Prompt for the Adult dataset**
>
> Given information about a person, you must predict if their income exceeds $50K/yr. Answer with one of the following: greater than 50K | less than or equal to 50K.
>
> **Example 1 -**
> **workclass**: Private , **hours per week**: 20, **sex**: Male, **age**: 17, **occupation**: Other-service, **capital loss**: 0, **education**: 10th, **capital gain**: 0, **marital status**: Never-married, **relationship**: Own-child, **Answer**: less than or equal to 50K
>
> **Example 2 -** .....
>
> **Workclass**: Private, **hours per week**: 40, **sex**: Female, **age**: 24, **occupation**: Sales, **capital loss**: 0, **education**: Some-college, **capital gain**: 0, **marital status**: Never-married, **relationship**: Own-child, **Answer**:

We take the negative of the language modelling loss (mean reduced cross-entropy) of each classification label with the language model output as the language model's un-normalized prediction **score** for that class ($SCORE_{LLM}$). Note that each classification label can contain multiple words. For example, 'Greater than 50K' and 'Less than or equal to 50K' for the Adult dataset. Thus, the loss calculation can be over a different number of tokens for each class, which is why we use mean

reduction. The exact loss extraction process will differ slightly depending on whether it is a Masked LLM or a Causal LLM. Finally, we center these raw scores around zero by subtracting the mean of the scores.

When combining GBDTs and TabPFN, it is straight forward to extract TabPFN scores as the model directly outputs un-normalized raw prediction scores.

## 3.2 BOOSTING LLM SCORES

Once we have the LLM scores, we use them to kickstart a GBDT. GBDT algorithms sequentially construct weak decision trees, where each tree is optimized to fit the residual error of the preceding ensemble. The input to the first tree is usually a constant value for all classes (Chen & Guestrin, 2016). Our approach is simply to replace this constant value with the LLM scores so that the GBDT algorithm learns the residual of the LLM prediction. This method can also be thought of as replacing the first tree in the ensemble with the LLM, during both training and inference. See Figure 1 for a representation of this procedure.

## 3.3 THE SCALING PARAMETER

We use a scaling parameter $s$ to scale the LLM scores before passing them on to the GBDT algorithm. By setting the scaling parameter to zero, our method is equivalent to the standalone GBDT; by making the scaling parameter very large, our method outputs predictions arbitrarily close to those of the LLM. In our experiments, we tune this hyper-parameter using Optuna (Akiba et al., 2019). We find that for intermediate values of this hyper-parameter, we can often achieve performance that exceeds both the GBDT and the LLM. Refer to Appendix E for an example.

The predictions of the ensemble consisting of the first $i$ trees are now

$$pred_{(0,i)} = pred_{(1,i)} + s * \text{SCORE}_{\text{LLM}} + C,$$

where $pred_{(a,b)}$ is the sum of the predictions of all the trees from $a$ to $b$; $s$ denotes the aforementioned scaling parameter that can take values $[0, \infty)$; $\text{SCORE}_{\text{LLM}}$ denotes the raw prediction of the LLM (or TabPFN) which is a vector in the case of classification (See 1); and $C$ denotes a constant which can be added to make $\text{SCORE}_{\text{LLM}}$ centered around 0 for numerical stability. Each tree i is progressively optimized so that $pred_{(0,i)}$ minimized following the standard gradient boosting procedure.

## 4 EXPERIMENTS AND ABLATIONS

Our primary experiments focus on boosting XGBoost (Chen & Guestrin, 2016) with Flan-T5-XXL (Chung et al., 2022) and TabPFN (Hollmann et al., 2023a) predictions. The Flan-T5-XXL model is the largest Flan-T5 variant with approximately 11 billion parameters. Flan-T5 models are created by multi-task instruction finetuning the standard T5 encoder-decoder model with chain-of-thought reasoning.

We additionally perform ablations on different GBDT and LLM model combinations as our boosting framework is agnostic with respect to both the precise LLM and GBDT. In addition to Flan-T5 models, we also include the newly released 8 billion paramter Meta-Llama-3-8B-Instruct (AI@Meta, 2024) model as a drop-in replacement. The Meta-Llama family is a collection of high performing decoder-only language models. Together, we conduct ablations including the GBDTs XG-Boost (Chen & Guestrin, 2016) and LightGBM (Ke et al., 2017), and including seeding mechanisms Flan-T5, Llama3-8B, and TabPFN.

### 4.1 DATASETS AND DATA PREPARATION

For our experiments, we adopt the UCI (Dua & Graff, 2017) datasets used by Slack & Singh (2023) together with the public tabular datasets used by Hegselmann et al. (2023) (TabLLM). We filter out the datasets which have more than 5 classes from the UCI datasets as few shot LLM performance is generally poor when the number of classes are high. The final 16 datasets used after filtering are listed in Table 1. As described in section 3.1 We prepare the data for few-shot (in-context) inference

utilizing the tools developed by Slack & Singh (2023). We sub-sample our datasets to much smaller sizes so that we have sufficient granularity to bridge the few-shot regime where LLMs/TabPFN excel at and the large dataset regime where GBDTs are better. We chose the sample sizes 10, 25, 50, 100, 200 and 500 for applicable datasets in addition to running the experiments on the full dataset.

Table 1: **The 16 Datasets used for our Experiments**

| UCI | TabLLM |
|---|---|
| Abalone | Bank |
| Adult (Also used for TabLLM) | Blood |
| Breast Cancer Wisconsin - Diagnostic | California |
| Churn | Car |
| Heart Disease | Credit-g |
| Shark Tank | Diabetes |
| Statlog - Australian Credit Approval | Heart |
| Wine | Jungle |

## 4.2 HYPERPARAMETER OPTIMIZATION

One of the benefits of using GBDT based methods is the ability to perform many rounds of Hyperparameter optimization (HPO) with a low computation budget. HPO is well known to increase performance of tabular models (Gorishniy et al., 2021) and is often included as part of the GBDT pipeline. We perform HPO using Optuna (Akiba et al., 2019). We use separate validation folds so that test data is new used for HPO trials. The hyper-parameter search spaces used for our GBDT experiments are listed in Appendix A.

For best results, we find that the scaling hyper-parameter $s$ should be independently tuned after tuning the standard GBDT hyper-parameters. We find that this makes the tuning process more stable and guarantees improvement in validation loss when including scaling. We tune the GBDT hyper-parameters for 100 Optuna trials and tune the scaling parameter for an additional 30 trials. Importantly, for our other ensembling baselines stated in section 4.4 we tune the GBDT hyper-parameters for 130 Optuna trials to keep the total HPO trials consistent.

## 4.3 COMPUTE RESOURCES

A major advantage of LLM-Boost is its lightweight overhead. The computational resources required for our boosting process, disregarding LLM costs, is the same as that required for HPO of GBDTs. Specifically, the boosting process can be performed on CPU. Full hyper-parameter tuning only takes up to 4 hours for the largest datasets on CPU. For few-shot LLM inference (Flan-T5-XXL and Meta-Llama-3-8B-Instruct) we use 4 RTX A4000 GPUs. Inference on the largest datasets we tested takes up to 18 hours to precompute. Importantly, this significantly less resource intensive compared to supervised fine tuning of LLMs for tabular tasks.

## 4.4 BASELINES

To validate our method, we first consider selecting the raw **LLM** and **GBDT** models as baselines. However, on average LLM-Boost performs much better than either the GBDT or the LLM model alone. Therefore, we utilize two strong and widely-used ensembling baselines and compare our LLM-Boost against them. The first baseline is **Selection**, i.e., selecting the best performing model out of the GBDT and LLM based on validation performance. The other is **Stacking**, i.e., appending LLM scores as additional features for GBDTs.

## 5 RESULTS

In this section, we only present the aggregate performance statistics detailed next for brevity and straightforward comparison between methods. Please see Appendix B for detailed results for each combination of models and datasets. We calculate the rank and z-score between the three methods

for each dataset at each train sample size based on AUC. Then, we average the rank and z-score across datasets of a given sample size to illustrate the variation in relative performance between the three methods across training sample sizes.

Average rank is an intuitive metric that is common in the tabular domain. Naturally, a lower value on this metric is better. However, it is a coarse metric because some models may obtain similar performance across all datasets and yet have very different average ranks. On the other hand, averaging AUC across datasets conveys the magnitudes by which one model outperforms another, but this metric can be dominated by a small number of datasets where the performance across models has a high variance. The average z-score metric described below mitigates this effect.

The z-score for a model on a single dataset is calculated as $z = \frac{a-\mu}{\sigma}$, the number of standard deviations a model's performance is away from the mean computed across all methods considered in that experience. A negative z-score implies that the given method's performance is below the mean of all methods. A higher positive z-score implies better performance. We then average a single model's z-scores over all datasets and obtain an average z-score for that model. We include average rank results in Appendix C, and we instead focus on average AUC and average z-score in the main body.

*Important Note:* Some of the datasets we use for our experiments have less than 250 samples. Therefore, results we show on dataset sizes larger than 250 are for a subset of these datasets. A detailed list of experiments for each dataset and size are included in the Appendix B.

## 5.1 XGBOOST + FLAN-T5-XXL

Figure 2 showcases our XGBoost + Flan-T5 results. Average z-score, rank and AUC are obtained by taking the mean of the row-wise z-score, rank and AUC across all 16 datasets at each given sample size shown in our full results Table 3. Each experiment in our full results is averaged over 5 seeds, and we show standard errors. We provide more details in Appendix B. As stated in Section 4, the model selection baseline is based on the validation performance of each model. All final XGBoost, stacking and LLM-Boost results are obtained after HPO.

As seen in the results graphs, LLM-Boost wins on all of the sample sizes both in terms of average rank and average z-score. We further reiterate that LLM-Boost significantly outperforms each of the stand-alone models.

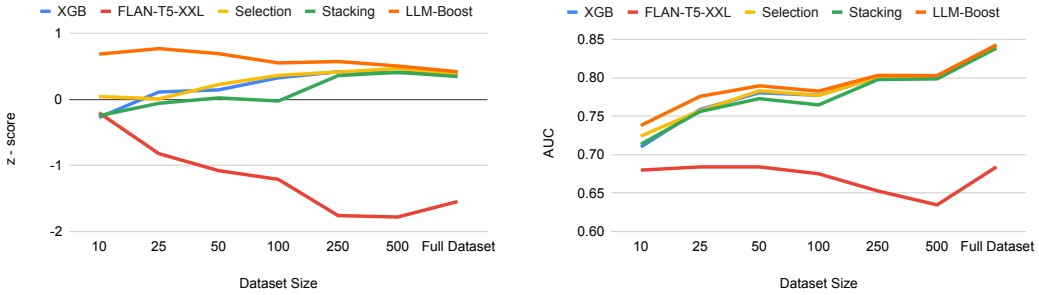

Figure 2: **LLM-Boost, combining Flan-T5-XXL and XGBoost, outperforms ensemble baselines and the standalone constituent models across dataset sizes.** Left: Average z-score based on AUC performance across dataset sizes for LLM-Boost and other ensemble baselines. Right: AUC performance across dataset sizes. *Important Note:* For this experiment we always compute the LLM Scores using a 3-shot prompt therefore the LLM performance remains constant throughout all trainset sizes where the extra data is only used for GBDT training. The trough in LLM performance in the 100-500 trainset range is due to us using only a subset of datasets which have sufficient training samples, for these data points.

## 5.2 XGBOOST + TABPFN

Figure 3 showcases the exceptionally strong performance of XGBoost + TabPFN in our experiments across all sample sizes. We note that the comparatively stronger performance of LLM-Boost in the XGBoost + TabPFN experiments may be a consequence of the TabPFN performance being much stronger than our LLM baselines. Since the learning mechanisms for TabPFN and XGBoost are quite different and they both have similar performance, it would be easier for them to learn useful tabular features complementary to each other. The standalone models learning complementary features will not benefit the selection baseline.

Figure 4 gives a direct AUC comparison between boosted and non-boosted Flan-T5-XXL and TabPFN on our datasets. Our experiments makes it clear that LLM-Boost with TabPFN yields better results except in the smallest dataset sizes. This is expected as TabPFN can use upto 1000 in-context examples, while the LLM can use far less. Although using a stronger or fine-tuned LLM might result in better performance, we conclude that using LLM-Boost with TabPFN+XGBoost is better suited on instances where data sample size is not severely limited. Our full results are given in Table 6 where each experiment is averaged over 5 seeds. For the full-dataset experiments where the train size is greater than 1000 we randomly select 1000 samples for TabPFN training which is the standard procedure followed in the original work.

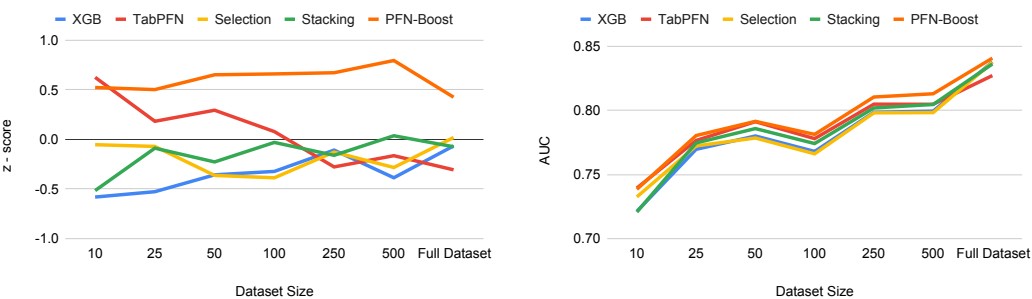

Figure 3: **Performance of LLM-Boost with TabPFN and XGBoost compared to the ensemble baselines and standalone models** Left:Average Z-Score based on AUC performance across dataset sizes for LLM-Boost and other ensemble baselines. Right: AUC across dataset sizes.

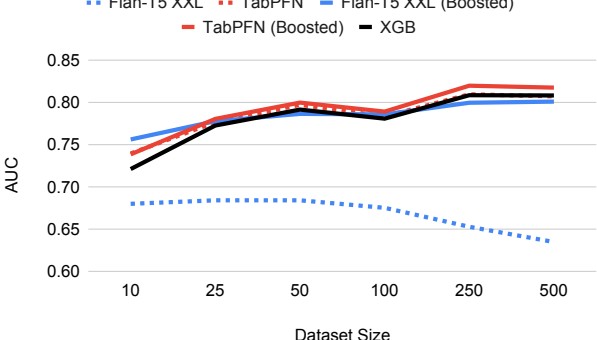

Figure 4: **Direct comparison of LLM-Boost with XGB+Flan-T5-XXL and XGB+TabPFN** We observe from this comparison that boosted TabPFN results are better except for the smallest dataset size. This is as expected as standalone TabPFN results are far superior to our standalone LLM results on average.

## 5.3 OTHER BOOSTING COMBINATIONS

We perform further LLM-Boost experiments with XGBoost+Llama-3-8B-Instruct and LightGBM+Flan-T5-XXL, in order to study sensitivity of LLM-Boost to the choice of language model and GBDT algorithm. Our XGB+Llama-3 experiments can be found in Figure 9. We find it more difficult to design prompts for the Llama-3-8B model to predict exactly the class label consistently compared to the Flan-T5 model. Therefore, the performance of the Llama-3 model is comparatively lower leading to lower boosted performance gains as well. The LightGBM experiments yield superior results compared to baselines in the small dataset sizes as seen in Figure 10. However, the performance gain for LLM-Boost is not as pronounced compared to the XGBoost experiments.

## 5.4 ABLATING THE VALUE OF COLUMN HEADERS BY SHUFFLING THEM

LLMs perform well in the few-shot tabular setting as they are able to make use of the column headers (column names), which are valuable metadata that traditional tabular models cannot parse. To investigate the importance of meaningful column headers for LLM-Boost, we conduct an experiment where we randomly shuffle the column headers between columns and compare performance degradation. Once the column headers are shuffled, all semantic meaning of a column disappears because it is no longer corresponding to the appropriate value. We conduct this experiment on the Adult dataset and we provide our boosted/standalone performance for both shuffled and direct column headers in Figure 5. We see there that the column headers are especially useful when the dataset size is small, yet the LLM provides an advantage over XGBoost alone for very small dataset sizes, even when the column headers are shuffled. As the dataset size grows, eventually all models perform comparably well.

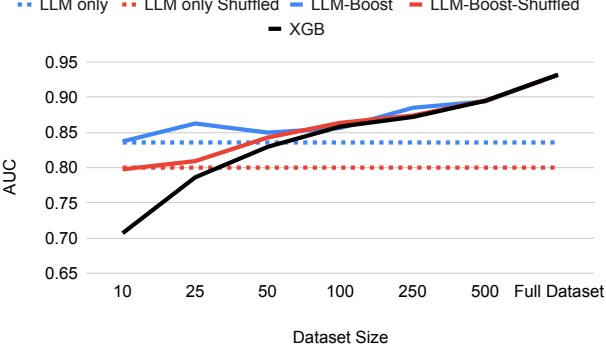

Figure 5: **This graph demonstrates the change in performance when the column headers are randomly shuffled.** To emulate datasets where column headers contain little or no semantic meaning we shuffle the column headers of the Adult Income Dataset and run LLM-Boost. It can be seen that even with the suffled column headers the LLM does provide some performance improvement over XGBoost when the datasize is small. However, LLM-Boost with meaningful column performs much better.

## 5.5 MODEL SIZE AND NUMBER OF FEW-SHOT SAMPLES

To test out whether the performance of our method is sensitive to the raw performance of the LLM, we conduct several experiments to analyze the impact of the LLM model size and the number of few-shot samples included in the LLM's prompt. These results can be found in Figure 6. We use different model sizes from the Flan-T5 model family to maintain as close as possible to an apples to apples comparison as different models might perform differently depending on how the prompt is engineered. As expected, both boosted and raw performance are best for the largest model when the most in-context examples (shots) are given. We do not experiment with higher number of shots since we run into context length limits for certain datasets.

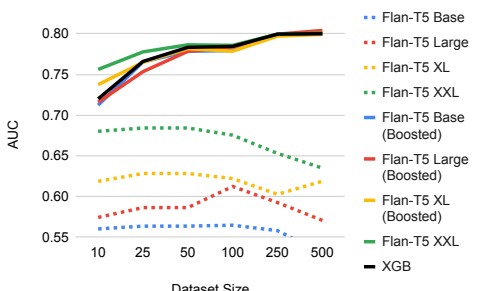 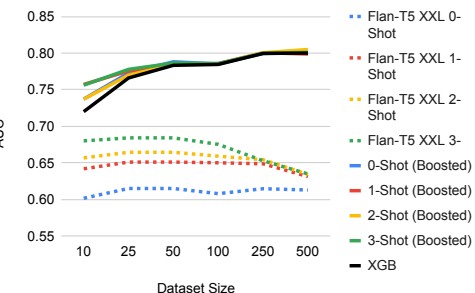

Figure 6: The graph on the left illustrates the change in **LLM-Boost performance with model size.** The graph on the right showcases the change in **LLM-Boost performance with varying number of few-shots**. Boosted and standalone LLM performance is included for both experiments

## 6 DISCUSSION AND CONCLUSION

In this paper, we show how to combine the benefits of LLMs and GDBTs for data-efficient predictions on tabular datasets. LLM-Boost often outperforms LLMs and GDBTs individually as well as ensemble baselines, adaptively focusing on the strengths of each method. In this final section, we close by discussing the limitations of our work as well as promising directions for future research.

**When to use LLM-Boost?** LLM-Boost showcases competitive performance on classification datasets with small to medium size training sets. In order to optimally benefit from fusing a language model to decision trees, we need semantically meaningful column headers. When such column headers are unavailable or the dataset is medium-sized or large, our variant combining TabPFN with GBDTs is highly effective.

**Limitations.** While we present promising performance on a slice of tabular datasets, we enumerate several limitations:

- The biggest drawback that restricts LLM-Boost is the necessity for interpretable text descriptors as column headers, namely column headers from which LLMs can extract meaning. Accordingly, some datasets may require prompt engineering.

- Language models are big in parameter count and slow, and they require GPUs for large-scale use. During training, LLM outputs can be pre-computed and re-used across all GBDT training runs with various hyperparameter configurations. After this one-time cost, training is no more expensive than GBDT training. For very large datasets, pre-computing LLM outputs may become a non-trivial cost.

- Common GBDT libraries are implemented in C++, accompanied by APIs in other languages such as Python. Maintaining high-speed training while simultaneously modifying the code for LLM-Boost may require implementing LLM-Boost in the original C++.

**Future work.** We finally present several promising directions for future research:

- Data scientists interact with tabular datasets, analyzing variable names, for example to engineer new features, and employing tools such as gradient-boosted decision trees. Our work is a first step towards automating this predictive modeling pipeline. A next step is to expand the capabilities of LLMs, for a full stack of data science functionality such as data visualization, hypothesis testing, and even suggesting valuable features to use.

- We only use three-shot prompting for the language model, but long-context methods may unlock the ability to feed far more training samples into the LLM. This possibility raises the question, will we still need XGBoost as LLMs gain the capability to ingest more data?

- Our boosting mechanism is model agnostic and may be expanded with other high performing tabular architectures such as Tabnet (Arik & Pfister, 2019) SAINT (Somepalli et al., 2021), NODE (Popov et al., 2019) and FT-Transformer (Gorishniy et al., 2021) in addition to LLMs.

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

## A  HYPER-PARAMETER SEARCH SPACES

Table 2: Hyper-parameter search spaces for our XGBoost and LightGBM experiments.

| XGBoost | |
|---|---|
| Parameter | Distribution |
| Max depth | $\text{UniformInt}[3, 10]$ |
| Min child weight | $\text{LogUniform}[1e\text{-}8, 1e5]$ |
| Subsample | $\text{Uniform}[0.5, 1]$ |
| Learning rate | $\text{LogUniform}[1e\text{-}5, 1]$ |
| Col sample by level | $\text{Uniform}[0.5, 1]$ |
| Col sample by tree | $\text{Uniform}[0.5, 1]$ |
| Gamma | $\{0, \text{LogUniform}[1e\text{-}8, 1e2]\}$ |
| Lambda | $\{0, \text{LogUniform}[1e\text{-}8, 1e2]\}$ |
| Alpha | $\{0, \text{LogUniform}[1e\text{-}8, 1e2]\}$ |
| Num boost rounds | 20 |
| **Scale** | $\{0, \text{LogUniform}[1e\text{-}4, 1e4]\}$ |
| # Iterations | 100 |

| LightGBM | |
|---|---|
| Parameter | Distribution |
| Num leaves | $\text{UniformInt}[2, 256]$ |
| Feature fraction | $\text{Uniform}[0.4, 1]$ |
| Bagging fraction | $\text{Uniform}[0.4, 1]$ |
| Bagging frequency | $\text{UniformInt}[1, 7]$ |
| Min child samples | $\text{UniformInt}[5, 100]$ |
| Lambda L1 | $\{0, \text{LogUniform}[1e\text{-}8, 10]\}$ |
| Lambda L2 | $\{0, \text{LogUniform}[1e\text{-}8, 10]\}$ |
| Num boost rounds | 100 |
| **Scale** | $\{0, \text{LogUniform}[1e\text{-}4, 1e4]\}$ |
| # Iterations | 100 |

## B  FULL RESULTS

Full experimental results can be found here. Each AUC result is obtained after 130 rounds of HPO. For LLM-Boost we perform HPO on the GBDT parameters for 100 Optuna (Akiba et al., 2019) trials followd by an additional 30 trias for the scaling paramater. For the selection and stacking baselines we perform HPO for 130 Optuna trials on the GBDT parameters. For each experiment each we randomly sub-sample train/val splits as well as sample HPO initilization over 5 different seeds and report mean AUC with standard error. However, we do not perform LLM inference over multiple few-shot train samples due to computational costs. Summarised Averaged row-wise rank and z-score metrics as well as Average AUC is given in C.

Table 3: Full accuracy results for our XGBoost + Flan-T5-XXL experiments

| Dataset | Train size | Val size | Test size | XGB ± Error | LLM | Selection (Best XGB/LLM) | Stacking ± Error | LLM-Boost ± Error (Ours) |
|---|---|---|---|---|---|---|---|---|
| Abalone | 10 | 10 | 836 | 0.6439 ± 0.0407 | 0.7528 | 0.6734 | 0.6438 ± 0.0372 | **0.6753 ± 0.0496** |
| | 25 | 25 | 836 | 0.6996 ± 0.0259 | 0.7528 | 0.7033 | 0.6976 ± 0.0235 | **0.7250 ± 0.0071** |
| | 50 | 50 | 836 | 0.7534 ± 0.0047 | 0.7528 | 0.7257 | 0.7515 ± 0.0041 | **0.7638 ± 0.0054** |
| | 100 | 100 | 836 | 0.7757 ± 0.0082 | 0.7528 | 0.7779 | **0.7802 ± 0.0067** | 0.7761 ± 0.0069 |
| | 250 | 250 | 836 | 0.8136 ± 0.0006 | 0.7528 | **0.8146** | 0.8099 ± 0.0016 | 0.8142 ± 0.0004 |
| | 500 | 500 | 836 | 0.8280 ± 0.0014 | 0.7528 | **0.8286** | 0.8224 ± 0.0012 | 0.8284 ± 0.0012 |
| | 1336 | 334 | 836 | 0.8469 ± 0.0003 | 0.7528 | 0.8468 | 0.8406 ± 0.0014 | **0.8469 ± 0.0005** |
| Adult | 10 | 10 | 1000 | 0.6998 ± 0.0119 | 0.8058 | 0.7411 | 0.7253 ± 0.0235 | **0.8093 ± 0.0078** |
| | 25 | 25 | 1000 | 0.7764 ± 0.0046 | 0.8058 | 0.7719 | 0.8177 ± 0.0191 | **0.8413 ± 0.0075** |
| | 50 | 50 | 1000 | 0.7997 ± 0.0085 | 0.8058 | 0.8088 | 0.8271 ± 0.0109 | **0.8431 ± 0.0056** |
| | 100 | 100 | 1000 | 0.8430 ± 0.0062 | 0.8058 | 0.8427 | 0.8486 ± 0.0033 | **0.8671 ± 0.0026** |
| | 250 | 250 | 1000 | 0.8680 ± 0.0019 | 0.8058 | 0.8683 | 0.8739 ± 0.0030 | **0.8796 ± 0.0011** |
| | 500 | 500 | 1000 | 0.8874 ± 0.0007 | 0.8058 | 0.8888 | 0.8885 ± 0.0019 | **0.8931 ± 0.0010** |
| | 15628 | 3907 | 1000 | 0.9313 ± 0.0006 | 0.8058 | **0.9318** | 0.9314 ± 0.0004 | 0.9314 ± 0.0005 |
| BreastCancer | 10 | 10 | 114 | 0.9789 ± 0.0031 | 0.9721 | 0.9762 | 0.9790 ± 0.0034 | **0.9822 ± 0.0019** |
| | 25 | 25 | 114 | 0.9804 ± 0.0016 | 0.9721 | 0.9806 | 0.9789 ± 0.0033 | **0.9829 ± 0.0020** |
| | 50 | 50 | 114 | 0.9783 ± 0.0015 | 0.9721 | 0.9777 | 0.9756 ± 0.0057 | **0.9792 ± 0.0016** |
| | 100 | 100 | 114 | 0.9807 ± 0.0023 | 0.9721 | 0.9802 | 0.9803 ± 0.0025 | **0.9822 ± 0.0021** |
| | 181 | 45 | 114 | 0.9871 ± 0.0009 | 0.9721 | 0.9864 | 0.9823 ± 0.0020 | **0.9871 ± 0.0009** |
| Churn | 10 | 10 | 1000 | 0.6848 ± 0.0206 | 0.7155 | 0.7110 | 0.6784 ± 0.0271 | **0.7257 ± 0.0189** |
| | 25 | 25 | 1000 | 0.7614 ± 0.0111 | 0.7155 | 0.7634 | 0.7558 ± 0.0120 | **0.7730 ± 0.0108** |
| | 50 | 50 | 1000 | 0.7868 ± 0.0047 | 0.7155 | 0.7831 | 0.7730 ± 0.0038 | **0.7906 ± 0.0074** |
| | 100 | 100 | 1000 | 0.7903 ± 0.0048 | 0.7155 | **0.7956** | 0.7934 ± 0.0055 | 0.7928 ± 0.0052 |
| | 250 | 250 | 1000 | 0.8076 ± 0.0020 | 0.7155 | **0.8083** | 0.8076 ± 0.0017 | 0.8082 ± 0.0021 |
| | 500 | 500 | 1000 | 0.8180 ± 0.0005 | 0.7155 | 0.8171 | 0.8122 ± 0.0024 | **0.8188 ± 0.0008** |
| | 2253 | 563 | 1000 | 0.8278 ± 0.0005 | 0.7155 | 0.8269 | 0.8237 ± 0.0020 | **0.8275 ± 0.0006** |
| HeartDisease | 10 | 10 | 61 | 0.8195 ± 0.0161 | 0.8621 | 0.8195 | **0.8362 ± 0.0090** | 0.8259 ± 0.0139 |
| | 25 | 25 | 61 | 0.8855 ± 0.0089 | 0.8621 | 0.8812 | 0.8964 ± 0.0068 | **0.8994 ± 0.0033** |
| | 50 | 50 | 61 | 0.9116 ± 0.0081 | 0.8621 | 0.9117 | **0.9219 ± 0.0026** | 0.9206 ± 0.0044 |
| | 96 | 24 | 61 | 0.9372 ± 0.0040 | 0.8621 | 0.9386 | **0.9499 ± 0.0018** | 0.9382 ± 0.0034 |
| Sharktank | 10 | 10 | 99 | 0.2509 ± 0.0398 | 0.5540 | 0.4930 | 0.2504 ± 0.0406 | **0.5331 ± 0.0304** |
| | 25 | 25 | 99 | 0.5241 ± 0.0358 | 0.5540 | 0.5197 | 0.2596 ± 0.0331 | **0.5452 ± 0.0275** |
| | 50 | 50 | 99 | 0.5184 ± 0.0114 | 0.5540 | 0.5325 | 0.5160 ± 0.0205 | **0.5370 ± 0.0154** |
| | 100 | 100 | 99 | 0.4908 ± 0.0173 | 0.5540 | 0.4872 | 0.4832 ± 0.0081 | **0.5113 ± 0.0062** |
| | 316 | 79 | 99 | 0.5213 ± 0.0036 | 0.5540 | 0.5130 | 0.5164 ± 0.0048 | **0.5213 ± 0.0036** |
| Statlog | 10 | 10 | 138 | 0.8037 ± 0.0400 | 0.8330 | 0.8071 | 0.7950 ± 0.0350 | **0.8432 ± 0.0121** |
| | 25 | 25 | 138 | 0.9018 ± 0.0064 | 0.8330 | **0.9063** | 0.9031 ± 0.0054 | 0.9061 ± 0.0041 |
| | 50 | 50 | 138 | 0.9163 ± 0.0038 | 0.8330 | 0.9157 | 0.9091 ± 0.0065 | **0.9161 ± 0.0037** |
| | 100 | 100 | 138 | 0.9300 ± 0.0046 | 0.8330 | 0.9292 | 0.9280 ± 0.0036 | **0.9300 ± 0.0046** |
| | 250 | 250 | 138 | 0.9323 ± 0.0024 | 0.8330 | 0.9300 | 0.9281 ± 0.0027 | **0.9324 ± 0.0023** |
| | 446 | 111 | 138 | 0.9191 ± 0.0017 | 0.8330 | 0.9204 | **0.9231 ± 0.0024** | 0.9193 ± 0.0017 |
| Wine | 10 | 10 | 36 | 0.9853 ± 0.0027 | 0.6304 | 0.9834 | **0.9854 ± 0.0027** | 0.9789 ± 0.0059 |
| | 25 | 25 | 36 | 0.9988 ± 0.0006 | 0.6304 | 0.9987 | 0.9980 ± 0.0009 | **0.9988 ± 0.0006** |
| | 50 | 50 | 36 | 0.9999 ± 0.0001 | 0.6304 | 0.9998 | 0.9998 ± 0.0001 | **0.9999 ± 0.0001** |
| | 113 | 28 | 36 | 0.9999 ± 0.0000 | 0.6304 | 1.0000 | **1.0000 ± 0.0000** | 0.9999 ± 0.0000 |
| Bank | 10 | 10 | 9043 | 0.5559 ± 0.0222 | 0.6915 | 0.6610 | 0.5599 ± 0.0323 | **0.6646 ± 0.0137** |
| | 25 | 25 | 9043 | 0.5968 ± 0.0424 | 0.6915 | 0.6022 | 0.6046 ± 0.0395 | **0.6446 ± 0.0573** |
| | 50 | 50 | 9043 | 0.6329 ± 0.0286 | 0.6915 | 0.6470 | 0.6578 ± 0.0303 | **0.6657 ± 0.0366** |
| | 100 | 100 | 9043 | 0.6756 ± 0.0211 | 0.6915 | 0.6752 | 0.6517 ± 0.0198 | **0.6836 ± 0.0192** |
| | 250 | 250 | 9043 | 0.7232 ± 0.0171 | 0.6915 | 0.7208 | 0.7175 ± 0.0138 | **0.7419 ± 0.0194** |
| | 500 | 500 | 9043 | 0.7392 ± 0.0114 | 0.6915 | **0.7431** | 0.7252 ± 0.0118 | 0.7257 ± 0.0098 |
| | 28934 | 7233 | 9043 | 0.7901 ± 0.0008 | 0.6915 | 0.7830 | 0.7863 ± 0.0016 | **0.7905 ± 0.0010** |
| Blood | 10 | 10 | 150 | 0.5220 ± 0.0115 | 0.5113 | 0.5207 | 0.5159 ± 0.0076 | **0.5236 ± 0.0117** |
| | 25 | 25 | 150 | 0.5205 ± 0.0283 | 0.5113 | 0.5142 | 0.4989 ± 0.0228 | **0.5262 ± 0.0294** |
| | 50 | 50 | 150 | 0.5268 ± 0.0259 | 0.5113 | 0.5289 | 0.5242 ± 0.0198 | **0.5300 ± 0.0262** |
| | 100 | 100 | 150 | 0.5295 ± 0.0138 | 0.5113 | **0.5362** | 0.5172 ± 0.0076 | 0.5295 ± 0.0138 |
| | 250 | 250 | 150 | 0.5399 ± 0.0079 | 0.5113 | 0.5404 | 0.5425 ± 0.0032 | **0.5449 ± 0.0067** |
| | 478 | 119 | 150 | 0.5343 ± 0.0022 | 0.5113 | 0.5399 | **0.5492 ± 0.0045** | 0.5380 ± 0.0009 |
| CalHousing | 10 | 10 | 4128 | 0.7490 ± 0.0308 | 0.7972 | 0.7844 | 0.7634 ± 0.0193 | **0.7965 ± 0.0225** |
| | 25 | 25 | 4128 | 0.8278 ± 0.0057 | 0.7972 | 0.8193 | 0.8269 ± 0.0037 | **0.8289 ± 0.0064** |
| | 50 | 50 | 4128 | 0.8370 ± 0.0102 | 0.7972 | 0.8310 | 0.8340 ± 0.0102 | **0.8405 ± 0.0134** |
| | 100 | 100 | 4128 | 0.8623 ± 0.0040 | 0.7972 | **0.8640** | 0.8571 ± 0.0020 | 0.8635 ± 0.0038 |
| | 250 | 250 | 4128 | 0.8853 ± 0.0026 | 0.7972 | **0.8864** | 0.8797 ± 0.0044 | 0.8862 ± 0.0033 |
| | 500 | 500 | 4128 | 0.8971 ± 0.0039 | 0.7972 | 0.8970 | 0.8897 ± 0.0066 | **0.8983 ± 0.0036** |
| | 13209 | 3302 | 4128 | 0.9198 ± 0.0010 | 0.7972 | 0.9164 | 0.9155 ± 0.0019 | **0.9203 ± 0.0008** |
| Car | 25 | 25 | 346 | 0.7266 ± 0.0097 | 0.7461 | 0.7146 | 0.6739 ± 0.0196 | **0.7892 ± 0.0141** |
| | 50 | 50 | 346 | 0.7750 ± 0.0085 | 0.7461 | 0.7799 | 0.6356 ± 0.0826 | **0.7958 ± 0.0142** |
| | 100 | 100 | 346 | 0.8354 ± 0.0076 | 0.7461 | 0.8189 | 0.7083 ± 0.0302 | **0.8567 ± 0.0085** |
| | 1089 | 272 | 346 | 0.8730 ± 0.0036 | 0.7461 | **0.8799** | 0.8390 ± 0.0024 | 0.8720 ± 0.0042 |
| Credit-g | 10 | 10 | 200 | 0.5845 ± 0.0289 | 0.2730 | 0.5827 | 0.5830 ± 0.0341 | **0.5845 ± 0.0289** |
| | 25 | 25 | 200 | 0.6451 ± 0.0146 | 0.2730 | 0.6467 | **0.6649 ± 0.0100** | 0.6398 ± 0.0167 |
| | 50 | 50 | 200 | 0.6641 ± 0.0153 | 0.2730 | 0.6695 | **0.6719 ± 0.0222** | 0.6641 ± 0.0153 |
| | 100 | 100 | 200 | 0.7088 ± 0.0123 | 0.2730 | 0.7124 | **0.7263 ± 0.0110** | 0.7088 ± 0.0123 |
| | 250 | 250 | 200 | 0.7420 ± 0.0081 | 0.2730 | 0.7460 | **0.7487 ± 0.0069** | 0.7393 ± 0.0090 |
| | 640 | 160 | 200 | 0.7777 ± 0.0017 | 0.2730 | 0.7757 | **0.7783 ± 0.0008** | 0.7780 ± 0.0019 |
| Diabetes | 10 | 10 | 154 | 0.6759 ± 0.0391 | 0.6386 | **0.6800** | 0.6691 ± 0.0412 | 0.6748 ± 0.0372 |
| | 25 | 25 | 154 | 0.7609 ± 0.0126 | 0.6386 | **0.7635** | 0.7411 ± 0.0137 | 0.7497 ± 0.0096 |
| | 50 | 50 | 154 | 0.7900 ± 0.0071 | 0.6386 | **0.7900** | 0.7743 ± 0.0062 | 0.7829 ± 0.0083 |
| | 100 | 100 | 154 | 0.8097 ± 0.0017 | 0.6386 | **0.8094** | 0.7900 ± 0.0075 | 0.8062 ± 0.0021 |
| | 250 | 250 | 154 | 0.8221 ± 0.0047 | 0.6386 | **0.8224** | 0.7951 ± 0.0034 | 0.8130 ± 0.0040 |
| | 491 | 122 | 154 | 0.8349 ± 0.0031 | 0.6386 | **0.8365** | 0.8170 ± 0.0040 | 0.8341 ± 0.0034 |
| Heart | 10 | 10 | 184 | 0.7910 ± 0.0129 | 0.5955 | **0.7930** | 0.7828 ± 0.0147 | 0.7888 ± 0.0149 |
| | 25 | 25 | 184 | 0.8209 ± 0.0119 | 0.5955 | **0.8160** | 0.8088 ± 0.0149 | 0.8159 ± 0.0122 |
| | 50 | 50 | 184 | 0.8332 ± 0.0096 | 0.5955 | 0.8311 | 0.8266 ± 0.0086 | **0.8327 ± 0.0096** |
| | 100 | 100 | 184 | 0.8405 ± 0.0058 | 0.5955 | **0.8414** | 0.8398 ± 0.0078 | 0.8397 ± 0.0041 |
| | 250 | 250 | 184 | 0.8553 ± 0.0028 | 0.5955 | **0.8526** | 0.8426 ± 0.0028 | 0.8526 ± 0.0034 |
| | 587 | 146 | 184 | 0.8719 ± 0.0023 | 0.5955 | 0.8713 | 0.8609 ± 0.0035 | **0.8713 ± 0.0021** |
| Jungle | 10 | 10 | 8964 | 0.6551 ± 0.0243 | 0.5659 | 0.6352 | **0.6611 ± 0.0262** | 0.6372 ± 0.0237 |
| | 25 | 25 | 8964 | 0.7127 ± 0.0157 | 0.5659 | 0.7112 | **0.7213 ± 0.0129** | 0.7205 ± 0.0141 |
| | 50 | 50 | 8964 | 0.7643 ± 0.0078 | 0.5659 | **0.7696** | 0.7659 ± 0.0083 | 0.7690 ± 0.0078 |
| | 100 | 100 | 8964 | 0.8070 ± 0.0054 | 0.5659 | 0.8063 | 0.8004 ± 0.0032 | **0.8077 ± 0.0057** |
| | 250 | 250 | 8964 | 0.8182 ± 0.0056 | 0.5659 | 0.8204 | **0.8254 ± 0.0062** | 0.8174 ± 0.0034 |
| | 500 | 500 | 8964 | 0.8491 ± 0.0039 | 0.5659 | **0.8468** | 0.8378 ± 0.0029 | 0.8251 ± 0.0037 |
| | 28684 | 7171 | 8964 | 0.9038 ± 0.0031 | 0.5659 | **0.9082** | 0.8995 ± 0.0032 | 0.9040 ± 0.0029 |

Table 4: Full accuracy results for our XGBoost + Llama-3-8B-Instruct experiments

| Dataset | Train size | Val size | Test size | XGB ± Error | LLM | Selection (Best XGB/LLM) | Stacking ± Error | LLM-Boost ± Error (Ours) |
|---|---|---|---|---|---|---|---|---|
| Abalone | 10 | 10 | 836 | 0.6471 ± 0.0412 | 0.7294 | 0.6442 | 0.6437 ± 0.0515 | **0.6712 ± 0.0276** |
| | 25 | 25 | 836 | 0.6970 ± 0.0260 | 0.7294 | 0.7042 | 0.6929 ± 0.0245 | **0.7415 ± 0.0087** |
| | 50 | 50 | 836 | 0.7512 ± 0.0052 | 0.7294 | 0.7507 | 0.7303 ± 0.0097 | **0.7644 ± 0.0039** |
| | 100 | 100 | 836 | 0.7795 ± 0.0071 | 0.7294 | 0.7786 | 0.7595 ± 0.0126 | **0.7832 ± 0.0073** |
| | 250 | 250 | 836 | 0.8132 ± 0.0012 | 0.7294 | **0.8136** | 0.7939 ± 0.0039 | 0.8132 ± 0.0016 |
| | 500 | 500 | 836 | 0.8296 ± 0.0004 | 0.7294 | 0.8280 | 0.8127 ± 0.0026 | **0.8298 ± 0.0006** |
| | 1336 | 334 | 836 | 0.8466 ± 0.0004 | 0.7294 | **0.8468** | 0.8399 ± 0.0015 | 0.8465 ± 0.0003 |
| Adult | 10 | 10 | 1000 | 0.6882 ± 0.0176 | 0.7059 | 0.7030 | 0.7027 ± 0.0162 | **0.7303 ± 0.0149** |
| | 25 | 25 | 1000 | 0.7799 ± 0.0080 | 0.7059 | 0.7754 | 0.7864 ± 0.0120 | **0.7968 ± 0.0136** |
| | 50 | 50 | 1000 | 0.8116 ± 0.0108 | 0.7059 | **0.8113** | 0.8001 ± 0.0094 | 0.8062 ± 0.0133 |
| | 100 | 100 | 1000 | 0.8436 ± 0.0057 | 0.7059 | **0.8445** | 0.8310 ± 0.0071 | 0.8415 ± 0.0080 |
| | 250 | 250 | 1000 | 0.8702 ± 0.0024 | 0.7059 | **0.8706** | 0.8628 ± 0.0021 | 0.8698 ± 0.0024 |
| | 500 | 500 | 1000 | 0.8882 ± 0.0005 | 0.7059 | **0.8890** | 0.8805 ± 0.0015 | 0.8878 ± 0.0007 |
| | 15628 | 3907 | 1000 | 0.9316 ± 0.0003 | 0.7059 | 0.9311 | **0.9321 ± 0.0008** | 0.9316 ± 0.0006 |
| BreastCancer | 10 | 10 | 114 | 0.9753 ± 0.0051 | 0.9346 | **0.9799** | 0.9733 ± 0.0040 | 0.9767 ± 0.0051 |
| | 25 | 25 | 114 | 0.9802 ± 0.0015 | 0.9346 | 0.9775 | 0.9792 ± 0.0029 | **0.9832 ± 0.0023** |
| | 50 | 50 | 114 | 0.9791 ± 0.0015 | 0.9346 | 0.9733 | 0.9796 ± 0.0008 | **0.9808 ± 0.0008** |
| | 100 | 100 | 114 | 0.9801 ± 0.0018 | 0.9346 | 0.9800 | 0.9790 ± 0.0036 | **0.9824 ± 0.0020** |
| | 181 | 45 | 114 | 0.9859 ± 0.0005 | 0.9346 | 0.9873 | 0.9836 ± 0.0012 | **0.9884 ± 0.0003** |
| Churn | 10 | 10 | 1000 | 0.6789 ± 0.0259 | 0.5323 | 0.6824 | **0.6919 ± 0.0200** | 0.6802 ± 0.0261 |
| | 25 | 25 | 1000 | 0.7669 ± 0.0143 | 0.5323 | **0.7669** | 0.7549 ± 0.0100 | 0.7659 ± 0.0141 |
| | 50 | 50 | 1000 | 0.7826 ± 0.0048 | 0.5323 | **0.7861** | 0.7826 ± 0.0042 | 0.7802 ± 0.0064 |
| | 100 | 100 | 1000 | 0.7928 ± 0.0061 | 0.5323 | **0.7932** | 0.7874 ± 0.0041 | 0.7913 ± 0.0058 |
| | 250 | 250 | 1000 | 0.8093 ± 0.0018 | 0.5323 | **0.8107** | 0.8083 ± 0.0018 | 0.8091 ± 0.0019 |
| | 500 | 500 | 1000 | 0.8177 ± 0.0011 | 0.5323 | **0.8184** | 0.8133 ± 0.0015 | 0.8175 ± 0.0011 |
| | 2253 | 563 | 1000 | 0.8281 ± 0.0003 | 0.5323 | 0.8271 | 0.8261 ± 0.0015 | **0.8281 ± 0.0002** |
| HeartDisease | 10 | 10 | 61 | 0.8161 ± 0.0066 | 0.8685 | 0.8181 | 0.8242 ± 0.0117 | **0.8562 ± 0.0119** |
| | 25 | 25 | 61 | 0.8977 ± 0.0062 | 0.8685 | 0.8906 | 0.8989 ± 0.0126 | **0.9087 ± 0.0078** |
| | 50 | 50 | 61 | 0.9072 ± 0.0058 | 0.8685 | 0.9132 | **0.9198 ± 0.0055** | 0.9075 ± 0.0041 |
| | 96 | 24 | 61 | 0.9352 ± 0.0030 | 0.8685 | 0.9410 | **0.9433 ± 0.0024** | 0.9347 ± 0.0026 |
| Sharktank | 10 | 10 | 99 | 0.4938 ± 0.0401 | 0.4941 | **0.4949** | 0.4925 ± 0.0395 | 0.4648 ± 0.0254 |
| | 25 | 25 | 99 | 0.5236 ± 0.0338 | 0.4941 | **0.5235** | 0.5196 ± 0.0343 | 0.5181 ± 0.0286 |
| | 50 | 50 | 99 | 0.5310 ± 0.0141 | 0.4941 | 0.5183 | 0.5240 ± 0.0129 | **0.5310 ± 0.0150** |
| | 100 | 100 | 99 | 0.4940 ± 0.0154 | 0.4941 | **0.5094** | 0.4953 ± 0.0152 | 0.5022 ± 0.0032 |
| | 316 | 79 | 99 | 0.5149 ± 0.0057 | 0.4941 | 0.5159 | **0.5259 ± 0.0056** | 0.5132 ± 0.0062 |
| Statlog | 10 | 10 | 138 | 0.7966 ± 0.0404 | 0.6618 | **0.7999** | 0.7948 ± 0.0381 | 0.7891 ± 0.0387 |
| | 25 | 25 | 138 | 0.9056 ± 0.0057 | 0.6618 | 0.9022 | 0.8955 ± 0.0075 | **0.9058 ± 0.0057** |
| | 50 | 50 | 138 | 0.9157 ± 0.0021 | 0.6618 | 0.9144 | 0.9073 ± 0.0073 | **0.9157 ± 0.0021** |
| | 100 | 100 | 138 | 0.9270 ± 0.0058 | 0.6618 | 0.9264 | 0.9195 ± 0.0062 | **0.9276 ± 0.0060** |
| | 250 | 250 | 138 | 0.9283 ± 0.0020 | 0.6618 | 0.9293 | **0.9313 ± 0.0019** | 0.9283 ± 0.0019 |
| | 446 | 111 | 138 | 0.9246 ± 0.0006 | 0.6618 | 0.9244 | 0.9231 ± 0.0029 | **0.9246 ± 0.0006** |
| Wine | 10 | 10 | 36 | 0.9856 ± 0.0017 | 0.7416 | 0.9857 | 0.9840 ± 0.0029 | **0.9871 ± 0.0007** |
| | 25 | 25 | 36 | 0.9985 ± 0.0004 | 0.7416 | 0.9987 | 0.9983 ± 0.0007 | **0.9987 ± 0.0004** |
| | 50 | 50 | 36 | 0.9998 ± 0.0001 | 0.7416 | 0.9997 | 0.9997 ± 0.0002 | **0.9999 ± 0.0001** |
| | 113 | 28 | 36 | 1.0000 ± 0.0000 | 0.7416 | **1.0000** | 1.0000 ± 0.0000 | 1.0000 ± 0.0000 |
| Bank | 10 | 10 | 9043 | 0.5494 ± 0.0372 | 0.5867 | **0.5998** | 0.5518 ± 0.0339 | 0.5963 ± 0.0243 |
| | 25 | 25 | 9043 | 0.6081 ± 0.0279 | 0.5867 | **0.6252** | 0.5905 ± 0.0307 | 0.6142 ± 0.0225 |
| | 50 | 50 | 9043 | 0.6379 ± 0.0319 | 0.5867 | 0.6342 | 0.6196 ± 0.0310 | **0.6404 ± 0.0329** |
| | 100 | 100 | 9043 | 0.6415 ± 0.0168 | 0.5867 | **0.6631** | 0.6606 ± 0.0171 | 0.6601 ± 0.0187 |
| | 250 | 250 | 9043 | 0.7187 ± 0.0196 | 0.5867 | 0.7129 | **0.7232 ± 0.0152** | 0.7202 ± 0.0188 |
| | 500 | 500 | 9043 | 0.7471 ± 0.0147 | 0.5867 | 0.7435 | **0.7526 ± 0.0123** | 0.7476 ± 0.0145 |
| | 28934 | 7233 | 9043 | 0.7858 ± 0.0023 | 0.5867 | 0.7792 | **0.7859 ± 0.0035** | 0.7856 ± 0.0025 |
| Blood | 10 | 10 | 150 | 0.5223 ± 0.0120 | 0.5039 | **0.5271** | 0.5152 ± 0.0138 | 0.5226 ± 0.0120 |
| | 25 | 25 | 150 | 0.5171 ± 0.0263 | 0.5039 | 0.5131 | 0.5181 ± 0.0261 | **0.5207 ± 0.0238** |
| | 50 | 50 | 150 | 0.5311 ± 0.0234 | 0.5039 | 0.5230 | 0.5289 ± 0.0226 | **0.5347 ± 0.0245** |
| | 100 | 100 | 150 | 0.5288 ± 0.0133 | 0.5039 | 0.5284 | **0.5301 ± 0.0140** | 0.5290 ± 0.0126 |
| | 250 | 250 | 150 | 0.5374 ± 0.0114 | 0.5039 | **0.5424** | 0.5404 ± 0.0053 | 0.5343 ± 0.0116 |
| | 478 | 119 | 150 | 0.5397 ± 0.0016 | 0.5039 | 0.5322 | 0.5243 ± 0.0058 | 0.5258 ± 0.0076 |
| CalHousing | 10 | 10 | 4128 | 0.7645 ± 0.0186 | 0.7076 | **0.7569** | 0.7542 ± 0.0229 | 0.7199 ± 0.0348 |
| | 25 | 25 | 4128 | 0.8269 ± 0.0062 | 0.7076 | **0.8281** | 0.8153 ± 0.0016 | 0.8101 ± 0.0129 |
| | 50 | 50 | 4128 | 0.8292 ± 0.0153 | 0.7076 | **0.8329** | 0.8231 ± 0.0105 | 0.8279 ± 0.0125 |
| | 100 | 100 | 4128 | 0.8650 ± 0.0052 | 0.7076 | 0.8622 | 0.8491 ± 0.0028 | **0.8652 ± 0.0053** |
| | 250 | 250 | 4128 | 0.8828 ± 0.0040 | 0.7076 | **0.8849** | 0.8692 ± 0.0055 | 0.8817 ± 0.0034 |
| | 500 | 500 | 4128 | 0.9001 ± 0.0047 | 0.7076 | **0.8986** | 0.8856 ± 0.0058 | 0.8985 ± 0.0043 |
| | 13209 | 3302 | 4128 | 0.9184 ± 0.0013 | 0.7076 | **0.9204** | 0.9101 ± 0.0034 | 0.9188 ± 0.0006 |
| Car | 25 | 25 | 346 | 0.7275 ± 0.0050 | 0.6760 | 0.7210 | 0.7279 ± 0.0291 | **0.7792 ± 0.0122** |
| | 50 | 50 | 346 | 0.7836 ± 0.0072 | 0.6760 | 0.7644 | 0.7598 ± 0.0421 | **0.8207 ± 0.0034** |
| | 100 | 100 | 346 | 0.8318 ± 0.0071 | 0.6760 | 0.8162 | 0.8361 ± 0.0097 | **0.8397 ± 0.0118** |
| | 1089 | 272 | 346 | 0.8760 ± 0.0018 | 0.6760 | 0.8720 | 0.8562 ± 0.0024 | **0.8803 ± 0.0025** |
| Credit-g | 10 | 10 | 200 | 0.5724 ± 0.0246 | 0.6317 | 0.5843 | 0.5778 ± 0.0284 | **0.6111 ± 0.0176** |
| | 25 | 25 | 200 | 0.6413 ± 0.0134 | 0.6317 | 0.6432 | 0.6442 ± 0.0151 | **0.6458 ± 0.0151** |
| | 50 | 50 | 200 | 0.6651 ± 0.0164 | 0.6317 | 0.6658 | 0.6631 ± 0.0171 | **0.6687 ± 0.0184** |
| | 100 | 100 | 200 | 0.7058 ± 0.0095 | 0.6317 | 0.7101 | 0.7013 ± 0.0128 | **0.7111 ± 0.0097** |
| | 250 | 250 | 200 | 0.7436 ± 0.0069 | 0.6317 | 0.7452 | 0.7379 ± 0.0088 | **0.7477 ± 0.0081** |
| | 640 | 160 | 200 | 0.7763 ± 0.0032 | 0.6317 | 0.7756 | 0.7717 ± 0.0029 | 0.7756 ± 0.0029 |
| Diabetes | 10 | 10 | 154 | 0.6846 ± 0.0369 | 0.8042 | 0.6856 | 0.6955 ± 0.0453 | **0.7700 ± 0.0250** |
| | 25 | 25 | 154 | 0.7704 ± 0.0101 | 0.8042 | 0.7587 | 0.7795 ± 0.0108 | **0.8081 ± 0.0065** |
| | 50 | 50 | 154 | 0.7867 ± 0.0086 | 0.8042 | 0.7890 | 0.8052 ± 0.0057 | **0.8076 ± 0.0117** |
| | 100 | 100 | 154 | 0.8109 ± 0.0034 | 0.8042 | 0.8082 | 0.8129 ± 0.0043 | **0.8254 ± 0.0033** |
| | 250 | 250 | 154 | 0.8232 ± 0.0052 | 0.8042 | 0.8249 | **0.8283 ± 0.0041** | 0.8277 ± 0.0033 |
| | 491 | 122 | 154 | 0.8316 ± 0.0015 | 0.8042 | 0.8329 | **0.8333 ± 0.0023** | 0.8328 ± 0.0020 |
| Heart | 10 | 10 | 184 | 0.7898 ± 0.0114 | 0.6521 | **0.7927** | 0.7849 ± 0.0136 | 0.7654 ± 0.0281 |
| | 25 | 25 | 184 | 0.8185 ± 0.0129 | 0.6521 | 0.8164 | 0.8125 ± 0.0086 | **0.8256 ± 0.0081** |
| | 50 | 50 | 184 | 0.8315 ± 0.0086 | 0.6521 | **0.8311** | 0.8268 ± 0.0077 | 0.8303 ± 0.0063 |
| | 100 | 100 | 184 | 0.8371 ± 0.0099 | 0.6521 | **0.8436** | 0.8274 ± 0.0034 | 0.8370 ± 0.0127 |
| | 250 | 250 | 184 | 0.8506 ± 0.0029 | 0.6521 | 0.8513 | 0.8415 ± 0.0045 | **0.8518 ± 0.0027** |
| | 587 | 146 | 184 | 0.8695 ± 0.0020 | 0.6521 | **0.8696** | 0.8536 ± 0.0037 | 0.8676 ± 0.0025 |
| Jungle | 10 | 10 | 8964 | 0.6495 ± 0.0272 | 0.4789 | 0.6506 | **0.6525 ± 0.0251** | 0.5869 ± 0.0415 |
| | 25 | 25 | 8964 | 0.7064 ± 0.0189 | 0.4789 | **0.7100** | 0.7016 ± 0.0182 | 0.6754 ± 0.0279 |
| | 50 | 50 | 8964 | 0.7622 ± 0.0057 | 0.4789 | **0.7661** | 0.7451 ± 0.0139 | 0.7577 ± 0.0071 |
| | 100 | 100 | 8964 | 0.8067 ± 0.0057 | 0.4789 | **0.8029** | 0.7930 ± 0.0057 | 0.7980 ± 0.0103 |
| | 250 | 250 | 8964 | 0.8215 ± 0.0059 | 0.4789 | **0.8249** | 0.8227 ± 0.0054 | 0.8217 ± 0.0057 |
| | 500 | 500 | 8964 | 0.8407 ± 0.0055 | 0.4789 | 0.8406 | 0.8358 ± 0.0041 | **0.8426 ± 0.0052** |
| | 28684 | 7171 | 8964 | 0.9096 ± 0.0032 | 0.4789 | 0.9050 | 0.8968 ± 0.0014 | **0.9087 ± 0.0036** |

Table 5: Full AUC results for our LightGBM + Flan-T5-XXL experiments

| Dataset | Train size | Val size | Test size | XGB ± Error | LLM | Selection (Best XGB/LLM) | Stacking ± Error | LLM-Boost ± Error (Ours) |
|---|---|---|---|---|---|---|---|---|
| Abalone | 10 | 10 | 836 | 0.5046 ± 0.0072 | 0.7528 | 0.7022 | 0.5177 ± 0.0173 | **0.7329 ± 0.0084** |
| | 25 | 25 | 836 | 0.7060 ± 0.0270 | 0.7528 | 0.7027 | 0.7021 ± 0.0176 | **0.7075 ± 0.0211** |
| | 50 | 50 | 836 | 0.7567 ± 0.0048 | 0.7528 | 0.7528 | 0.7473 ± 0.0043 | **0.7623 ± 0.0058** |
| | 100 | 100 | 836 | 0.7780 ± 0.0080 | 0.7528 | **0.7782** | 0.7776 ± 0.0067 | 0.7759 ± 0.0079 |
| | 250 | 250 | 836 | 0.8165 ± 0.0007 | 0.7528 | 0.8160 | 0.8108 ± 0.0011 | **0.8164 ± 0.0012** |
| | 500 | 500 | 836 | 0.8307 ± 0.0009 | 0.7528 | 0.8304 | 0.8264 ± 0.0016 | 0.8294 ± 0.0014 |
| | 1336 | 334 | 836 | 0.8489 ± 0.0006 | 0.7528 | 0.8491 | 0.8441 ± 0.0002 | 0.8486 ± 0.0005 |
| Adult | 10 | 10 | 1000 | 0.5515 ± 0.0245 | 0.8058 | 0.7694 | 0.5419 ± 0.0260 | **0.8070 ± 0.0013** |
| | 25 | 25 | 1000 | 0.7641 ± 0.0066 | 0.8058 | 0.7713 | 0.8296 ± 0.0054 | **0.8448 ± 0.0085** |
| | 50 | 50 | 1000 | 0.7415 ± 0.0095 | 0.8058 | 0.7493 | 0.8138 ± 0.0055 | **0.8334 ± 0.0049** |
| | 100 | 100 | 1000 | 0.7421 ± 0.0155 | 0.8058 | 0.7475 | 0.8168 ± 0.0013 | **0.8203 ± 0.0096** |
| | 250 | 250 | 1000 | 0.8456 ± 0.0068 | 0.8058 | 0.8540 | 0.8515 ± 0.0030 | **0.8595 ± 0.0021** |
| | 500 | 500 | 1000 | 0.8903 ± 0.0017 | 0.8058 | 0.8921 | 0.8829 ± 0.0021 | **0.8938 ± 0.0009** |
| | 15628 | 3907 | 1000 | 0.9336 ± 0.0001 | 0.8058 | 0.9337 | 0.9333 ± 0.0002 | **0.9338 ± 0.0001** |
| BreastCancer | 25 | 25 | 114 | 0.9803 ± 0.0012 | 0.9721 | 0.9785 | 0.9810 ± 0.0035 | **0.9829 ± 0.0017** |
| | 50 | 50 | 114 | 0.9757 ± 0.0038 | 0.9721 | 0.9785 | **0.9808 ± 0.0011** | 0.9798 ± 0.0021 |
| | 100 | 100 | 114 | 0.9788 ± 0.0025 | 0.9721 | 0.9799 | 0.9824 ± 0.0012 | **0.9827 ± 0.0017** |
| | 181 | 45 | 114 | 0.9873 ± 0.0006 | 0.9721 | 0.9874 | 0.9894 ± 0.0004 | **0.9895 ± 0.0008** |
| Churn | 10 | 10 | 1000 | 0.5548 ± 0.0341 | 0.7155 | **0.7155** | 0.5658 ± 0.0414 | 0.6968 ± 0.0200 |
| | 25 | 25 | 1000 | 0.7688 ± 0.0109 | 0.7155 | 0.7605 | 0.7559 ± 0.0140 | **0.7812 ± 0.0035** |
| | 50 | 50 | 1000 | 0.7833 ± 0.0066 | 0.7155 | 0.7841 | 0.7758 ± 0.0050 | **0.7876 ± 0.0036** |
| | 100 | 100 | 1000 | 0.7942 ± 0.0046 | 0.7155 | 0.7936 | 0.7914 ± 0.0040 | **0.7986 ± 0.0060** |
| | 250 | 250 | 1000 | 0.8062 ± 0.0021 | 0.7155 | 0.8079 | 0.8055 ± 0.0020 | **0.8082 ± 0.0019** |
| | 500 | 500 | 1000 | 0.8171 ± 0.0009 | 0.7155 | 0.8191 | 0.8156 ± 0.0007 | 0.8186 ± 0.0013 |
| | 2253 | 563 | 1000 | 0.8286 ± 0.0001 | 0.7155 | 0.8283 | 0.8266 ± 0.0005 | 0.8286 ± 0.0001 |
| HeartDisease | 10 | 10 | 61 | 0.6103 ± 0.0471 | 0.8621 | 0.7516 | 0.6481 ± 0.0414 | **0.8348 ± 0.0162** |
| | 25 | 25 | 61 | 0.8927 ± 0.0121 | 0.8621 | 0.8950 | **0.9015 ± 0.0108** | 0.8996 ± 0.0092 |
| | 50 | 50 | 61 | 0.9229 ± 0.0057 | 0.8621 | 0.9276 | 0.9283 ± 0.0043 | **0.9299 ± 0.0047** |
| | 96 | 24 | 61 | 0.9402 ± 0.0010 | 0.8621 | 0.9402 | **0.9488 ± 0.0009** | 0.9408 ± 0.0008 |
| Sharktank | 10 | 10 | 99 | 0.4986 ± 0.0033 | 0.5540 | 0.4978 | 0.5003 ± 0.0050 | **0.5356 ± 0.0109** |
| | 25 | 25 | 99 | 0.5238 ± 0.0300 | 0.5540 | 0.5264 | 0.5265 ± 0.0305 | **0.5395 ± 0.0194** |
| | 50 | 50 | 99 | 0.5240 ± 0.0122 | 0.5540 | 0.5326 | 0.5101 ± 0.0250 | **0.5524 ± 0.0162** |
| | 100 | 100 | 99 | 0.4907 ± 0.0245 | 0.5540 | **0.4949** | 0.4878 ± 0.0212 | 0.4935 ± 0.0254 |
| | 316 | 79 | 99 | 0.5348 ± 0.0014 | 0.5540 | **0.5364** | 0.5140 ± 0.0025 | 0.5348 ± 0.0014 |
| Statlog | 10 | 10 | 138 | 0.5448 ± 0.0448 | 0.8330 | 0.6602 | 0.6375 ± 0.0702 | **0.8240 ± 0.0090** |
| | 25 | 25 | 138 | 0.8812 ± 0.0130 | 0.8330 | **0.8923** | 0.8737 ± 0.0073 | 0.8826 ± 0.0129 |
| | 50 | 50 | 138 | 0.8924 ± 0.0056 | 0.8330 | **0.8998** | 0.8781 ± 0.0110 | 0.8932 ± 0.0062 |
| | 100 | 100 | 138 | 0.9146 ± 0.0052 | 0.8330 | 0.9090 | 0.9087 ± 0.0060 | **0.9145 ± 0.0052** |
| | 250 | 250 | 138 | 0.9224 ± 0.0028 | 0.8330 | **0.9240** | 0.9239 ± 0.0016 | 0.9223 ± 0.0027 |
| | 446 | 111 | 138 | 0.9147 ± 0.0010 | 0.8330 | 0.9162 | **0.9187 ± 0.0020** | 0.9144 ± 0.0009 |
| Wine | 10 | 10 | 36 | 0.5000 ± 0.0000 | 0.6304 | 0.6304 | 0.3865 ± 0.0000 | **0.6372 ± 0.0000** |
| | 25 | 25 | 36 | 0.9984 ± 0.0003 | 0.6304 | 0.9982 | 0.9978 ± 0.0007 | **0.9983 ± 0.0004** |
| | 50 | 50 | 36 | 0.9998 ± 0.0001 | 0.6304 | 0.9998 | **0.9998 ± 0.0001** | 0.9998 ± 0.0000 |
| | 113 | 28 | 36 | 1.0000 ± 0.0000 | 0.6304 | 1.0000 | 1.0000 ± 0.0000 | 1.0000 ± 0.0000 |
| Bank | 10 | 10 | 9043 | 0.5000 ± 0.0000 | 0.6915 | **0.6915** | 0.5000 ± 0.0000 | 0.6768 ± 0.0147 |
| | 25 | 25 | 9043 | 0.6004 ± 0.0354 | 0.6915 | **0.6591** | 0.6131 ± 0.0412 | 0.6561 ± 0.0061 |
| | 50 | 50 | 9043 | 0.6415 ± 0.0361 | 0.6915 | 0.6492 | 0.6570 ± 0.0383 | **0.6625 ± 0.0410** |
| | 100 | 100 | 9043 | 0.6967 ± 0.0237 | 0.6915 | **0.7125** | 0.6778 ± 0.0240 | 0.7040 ± 0.0210 |
| | 250 | 250 | 9043 | 0.7320 ± 0.0105 | 0.6915 | 0.7366 | **0.7609 ± 0.0096** | 0.7383 ± 0.0084 |
| | 500 | 500 | 9043 | 0.7638 ± 0.0040 | 0.6915 | 0.7640 | **0.7690 ± 0.0099** | 0.7688 ± 0.0055 |
| | 28934 | 7233 | 9043 | 0.7838 ± 0.0008 | 0.6915 | 0.7840 | **0.7879 ± 0.0009** | 0.7868 ± 0.0017 |
| Blood | 10 | 10 | 150 | 0.5334 ± 0.0080 | 0.5113 | 0.5216 | 0.5000 ± 0.0000 | **0.5407 ± 0.0153** |
| | 25 | 25 | 150 | 0.5154 ± 0.0224 | 0.5113 | **0.5263** | 0.4953 ± 0.0186 | 0.5167 ± 0.0217 |
| | 50 | 50 | 150 | 0.5184 ± 0.0204 | 0.5113 | **0.5287** | 0.5057 ± 0.0198 | 0.5256 ± 0.0206 |
| | 100 | 100 | 150 | 0.5362 ± 0.0119 | 0.5113 | 0.5308 | 0.5065 ± 0.0050 | **0.5404 ± 0.0112** |
| | 250 | 250 | 150 | 0.5392 ± 0.0122 | 0.5113 | 0.5384 | 0.5367 ± 0.0086 | **0.5416 ± 0.0123** |
| | 478 | 119 | 150 | 0.5410 ± 0.0040 | 0.5113 | 0.5329 | 0.5321 ± 0.0055 | **0.5457 ± 0.0035** |
| CalHousing | 25 | 25 | 4128 | 0.7990 ± 0.0077 | 0.7972 | 0.8013 | **0.8130 ± 0.0047** | 0.7906 ± 0.0186 |
| | 50 | 50 | 4128 | 0.8270 ± 0.0110 | 0.7972 | 0.8317 | **0.8359 ± 0.0063** | 0.8256 ± 0.0092 |
| | 100 | 100 | 4128 | 0.8647 ± 0.0041 | 0.7972 | 0.8608 | 0.8575 ± 0.0044 | **0.8640 ± 0.0042** |
| | 250 | 250 | 4128 | 0.8832 ± 0.0037 | 0.7972 | 0.8854 | **0.8864 ± 0.0029** | 0.8829 ± 0.0033 |
| | 500 | 500 | 4128 | 0.8991 ± 0.0040 | 0.7972 | **0.9011** | 0.9005 ± 0.0060 | 0.8997 ± 0.0041 |
| | 13209 | 3302 | 4128 | 0.9181 ± 0.0030 | 0.7972 | 0.9132 | **0.9229 ± 0.0006** | 0.9180 ± 0.0029 |
| Car | 25 | 25 | 346 | 0.7855 ± 0.0063 | 0.7461 | **0.7882** | 0.7468 ± 0.0106 | 0.7846 ± 0.0154 |
| | 50 | 50 | 346 | 0.8309 ± 0.0057 | 0.7461 | **0.8380** | 0.6972 ± 0.0152 | 0.8214 ± 0.0054 |
| | 100 | 100 | 346 | 0.8689 ± 0.0062 | 0.7461 | **0.8684** | 0.7914 ± 0.0122 | 0.8519 ± 0.0069 |
| | 1089 | 272 | 346 | 0.9096 ± 0.0040 | 0.7461 | **0.9102** | 0.8518 ± 0.0030 | 0.9095 ± 0.0036 |
| Credit-g | 10 | 10 | 200 | 0.5313 ± 0.0187 | 0.2730 | 0.4863 | **0.5463 ± 0.0194** | 0.4859 ± 0.0559 |
| | 25 | 25 | 200 | 0.6099 ± 0.0079 | 0.2730 | 0.6223 | **0.6353 ± 0.0157** | 0.6092 ± 0.0086 |
| | 50 | 50 | 200 | 0.5790 ± 0.0143 | 0.2730 | 0.5791 | **0.6270 ± 0.0107** | 0.5777 ± 0.0138 |
| | 100 | 100 | 200 | 0.6062 ± 0.0156 | 0.2730 | 0.6091 | **0.6848 ± 0.0029** | 0.6060 ± 0.0157 |
| | 250 | 250 | 200 | 0.7000 ± 0.0064 | 0.2730 | 0.7045 | **0.7223 ± 0.0053** | 0.6998 ± 0.0064 |
| | 640 | 160 | 200 | 0.7808 ± 0.0029 | 0.2730 | 0.7826 | **0.7829 ± 0.0029** | 0.7808 ± 0.0029 |
| Diabetes | 10 | 10 | 154 | 0.5270 ± 0.0081 | 0.6386 | **0.6386** | 0.5330 ± 0.0330 | 0.6192 ± 0.0145 |
| | 25 | 25 | 154 | 0.7530 ± 0.0056 | 0.6386 | **0.7649** | 0.7533 ± 0.0111 | 0.7427 ± 0.0144 |
| | 50 | 50 | 154 | 0.7811 ± 0.0048 | 0.6386 | 0.7787 | **0.7841 ± 0.0032** | 0.7801 ± 0.0049 |
| | 100 | 100 | 154 | 0.7965 ± 0.0075 | 0.6386 | **0.8012** | 0.7933 ± 0.0080 | 0.7975 ± 0.0069 |
| | 250 | 250 | 154 | 0.8281 ± 0.0058 | 0.6386 | **0.8254** | 0.8197 ± 0.0047 | 0.8144 ± 0.0052 |
| | 491 | 122 | 154 | 0.8428 ± 0.0009 | 0.6386 | **0.8419** | 0.8294 ± 0.0008 | 0.8287 ± 0.0013 |
| Heart | 10 | 10 | 184 | 0.5284 ± 0.0284 | 0.5955 | 0.5892 | **0.6102 ± 0.0282** | 0.6047 ± 0.0093 |
| | 25 | 25 | 184 | 0.8130 ± 0.0101 | 0.5955 | **0.8162** | 0.8005 ± 0.0070 | 0.8067 ± 0.0132 |
| | 50 | 50 | 184 | 0.8170 ± 0.0127 | 0.5955 | 0.8170 | **0.8177 ± 0.0110** | 0.8057 ± 0.0136 |
| | 100 | 100 | 184 | 0.8172 ± 0.0125 | 0.5955 | 0.8173 | **0.8182 ± 0.0091** | 0.8130 ± 0.0124 |
| | 250 | 250 | 184 | 0.8557 ± 0.0043 | 0.5955 | **0.8562** | 0.8526 ± 0.0031 | 0.8527 ± 0.0044 |
| | 587 | 146 | 184 | 0.8731 ± 0.0013 | 0.5955 | **0.8775** | 0.8759 ± 0.0006 | 0.8717 ± 0.0014 |
| Jungle | 10 | 10 | 8964 | 0.5492 ± 0.0294 | 0.5659 | 0.5762 | 0.5289 ± 0.0289 | **0.5920 ± 0.0206** |
| | 25 | 25 | 8964 | 0.7156 ± 0.0124 | 0.5659 | 0.6916 | **0.7163 ± 0.0145** | 0.6871 ± 0.0319 |
| | 50 | 50 | 8964 | 0.7636 ± 0.0061 | 0.5659 | **0.7718** | 0.7650 ± 0.0114 | 0.7683 ± 0.0063 |
| | 100 | 100 | 8964 | 0.8099 ± 0.0026 | 0.5659 | 0.8062 | 0.8028 ± 0.0042 | **0.8094 ± 0.0024** |
| | 250 | 250 | 8964 | 0.8259 ± 0.0046 | 0.5659 | 0.8257 | **0.8279 ± 0.0046** | 0.8259 ± 0.0047 |
| | 500 | 500 | 8964 | 0.8433 ± 0.0050 | 0.5659 | **0.8440** | 0.8436 ± 0.0029 | 0.8435 ± 0.0049 |
| | 28684 | 7171 | 8964 | 0.9123 ± 0.0015 | 0.5659 | 0.9105 | 0.9039 ± 0.0004 | **0.9126 ± 0.0008** |

Table 6: Full AUC results for our XGBoost + TabPFN experiments

| Dataset | Train size | Val size | Test size | XGB ± Error | LLM | Selection (Best XGB/LLM) | Stacking ± Error | LLM-Boost ± Error (Ours) |
|---|---|---|---|---|---|---|---|---|
| Abalone | 10 | 10 | 836 | 0.6757 ± 0.0040 | 0.7109 | 0.6929 | 0.6985 ± 0.0193 | **0.7119 ± 0.0309** |
| | 25 | 25 | 836 | 0.7028 ± 0.0241 | 0.7418 | 0.7028 | 0.7193 ± 0.0271 | **0.7438 ± 0.0214** |
| | 50 | 50 | 836 | 0.7525 ± 0.0042 | 0.8073 | 0.7492 | 0.7804 ± 0.0090 | **0.8067 ± 0.0040** |
| | 100 | 100 | 836 | 0.7813 ± 0.0067 | 0.8247 | 0.7786 | 0.8075 ± 0.0069 | **0.8216 ± 0.0050** |
| | 250 | 250 | 836 | 0.8133 ± 0.0013 | 0.8421 | 0.8132 | 0.8335 ± 0.0008 | **0.8403 ± 0.0014** |
| | 500 | 500 | 836 | 0.8277 ± 0.0013 | 0.8497 | 0.8285 | 0.8432 ± 0.0018 | **0.8491 ± 0.0009** |
| | 1336 | 334 | 836 | 0.8454 ± 0.0003 | 0.8531 | 0.8454 | 0.8545 ± 0.0003 | **0.8559 ± 0.0002** |
| Adult | 10 | 10 | 1000 | 0.6752 ± 0.0567 | 0.6321 | **0.6791** | 0.6575 ± 0.0217 | 0.6656 ± 0.0681 |
| | 25 | 25 | 1000 | 0.7541 ± 0.0080 | 0.7153 | **0.7563** | 0.7369 ± 0.0114 | 0.7491 ± 0.0140 |
| | 50 | 50 | 1000 | 0.7571 ± 0.0058 | 0.7593 | 0.7469 | **0.7750 ± 0.0108** | 0.7659 ± 0.0078 |
| | 100 | 100 | 1000 | 0.7990 ± 0.0070 | 0.7947 | 0.7958 | 0.7985 ± 0.0044 | **0.8099 ± 0.0054** |
| | 250 | 250 | 1000 | 0.8540 ± 0.0044 | 0.8262 | 0.8515 | 0.8388 ± 0.0016 | **0.8579 ± 0.0033** |
| | 500 | 500 | 1000 | 0.8800 ± 0.0057 | 0.8462 | 0.8816 | 0.8659 ± 0.0024 | **0.8820 ± 0.0048** |
| | 15628 | 3907 | 1000 | 0.9234 ± 0.0018 | 0.8646 | 0.9234 | 0.9238 ± 0.0012 | **0.9241 ± 0.0015** |
| BreastCancer | 10 | 10 | 114 | 0.9790 ± 0.0030 | 0.9865 | 0.9798 | 0.9788 ± 0.0042 | **0.9817 ± 0.0020** |
| | 25 | 25 | 114 | 0.9790 ± 0.0023 | 0.9892 | 0.9815 | 0.9812 ± 0.0026 | **0.9844 ± 0.0040** |
| | 50 | 50 | 114 | 0.9768 ± 0.0032 | 0.9882 | 0.9761 | 0.9765 ± 0.0046 | **0.9796 ± 0.0046** |
| | 100 | 100 | 114 | 0.9800 ± 0.0019 | 0.9909 | 0.9804 | 0.9822 ± 0.0022 | **0.9852 ± 0.0032** |
| | 181 | 45 | 114 | 0.9855 ± 0.0006 | 0.9930 | 0.9859 | 0.9877 ± 0.0017 | **0.9931 ± 0.0000** |
| Churn | 10 | 10 | 1000 | 0.7122 ± 0.0000 | 0.6958 | 0.7151 | **0.7168 ± 0.0000** | 0.7122 ± 0.0000 |
| | 25 | 25 | 1000 | 0.7663 ± 0.0116 | 0.7361 | 0.7658 | 0.7551 ± 0.0098 | **0.7685 ± 0.0117** |
| | 50 | 50 | 1000 | 0.7829 ± 0.0045 | 0.7612 | **0.7854** | 0.7836 ± 0.0048 | 0.7836 ± 0.0048 |
| | 100 | 100 | 1000 | 0.7944 ± 0.0043 | 0.7745 | 0.7941 | 0.7825 ± 0.0035 | **0.7951 ± 0.0045** |
| | 250 | 250 | 1000 | 0.8052 ± 0.0028 | 0.7992 | **0.8078** | 0.7994 ± 0.0027 | 0.8071 ± 0.0031 |
| | 500 | 500 | 1000 | 0.8165 ± 0.0009 | 0.8099 | 0.8168 | 0.8107 ± 0.0016 | **0.8180 ± 0.0011** |
| | 2253 | 563 | 1000 | 0.8282 ± 0.0005 | 0.8135 | 0.8278 | 0.8226 ± 0.0007 | **0.8281 ± 0.0005** |
| HeartDisease | 10 | 10 | 61 | 0.8303 ± 0.0128 | 0.8506 | 0.8358 | 0.8128 ± 0.0150 | **0.8362 ± 0.0171** |
| | 25 | 25 | 61 | 0.8883 ± 0.0085 | 0.9075 | 0.8908 | 0.8986 ± 0.0074 | **0.9018 ± 0.0078** |
| | 50 | 50 | 61 | 0.9158 ± 0.0050 | 0.9188 | 0.9181 | 0.9177 ± 0.0066 | **0.9215 ± 0.0036** |
| | 96 | 24 | 61 | 0.9399 ± 0.0032 | 0.9175 | **0.9423** | 0.9181 ± 0.0025 | 0.9397 ± 0.0028 |
| Sharktank | 10 | 10 | 99 | 0.5120 ± 0.0338 | 0.4984 | 0.5085 | 0.5017 ± 0.0340 | **0.5122 ± 0.0359** |
| | 25 | 25 | 99 | 0.5136 ± 0.0208 | 0.5046 | **0.5181** | 0.5027 ± 0.0214 | 0.5118 ± 0.0209 |
| | 50 | 50 | 99 | 0.5040 ± 0.0047 | 0.5241 | **0.5052** | 0.5022 ± 0.0022 | 0.5020 ± 0.0046 |
| | 100 | 100 | 99 | 0.4864 ± 0.0175 | 0.4964 | 0.4898 | **0.4988 ± 0.0123** | 0.4877 ± 0.0163 |
| | 316 | 79 | 99 | 0.4845 ± 0.0092 | 0.4564 | 0.4817 | **0.4864 ± 0.0134** | 0.4751 ± 0.0078 |
| Statlog | 10 | 10 | 138 | 0.8013 ± 0.0388 | 0.8212 | 0.8020 | 0.7946 ± 0.0358 | **0.8099 ± 0.0422** |
| | 25 | 25 | 138 | 0.9048 ± 0.0057 | 0.8891 | **0.9056** | 0.8975 ± 0.0063 | 0.9031 ± 0.0035 |
| | 50 | 50 | 138 | 0.9123 ± 0.0029 | 0.9011 | **0.9150** | 0.9080 ± 0.0065 | 0.9138 ± 0.0031 |
| | 100 | 100 | 138 | 0.9274 ± 0.0041 | 0.9135 | 0.9262 | 0.9134 ± 0.0068 | **0.9264 ± 0.0047** |
| | 250 | 250 | 138 | 0.9319 ± 0.0021 | 0.9195 | **0.9297** | 0.9187 ± 0.0025 | 0.9267 ± 0.0014 |
| | 446 | 111 | 138 | 0.9247 ± 0.0012 | 0.9178 | 0.9237 | 0.9177 ± 0.0007 | **0.9253 ± 0.0007** |
| Wine | 10 | 10 | 36 | 0.9881 ± 0.0004 | 0.9923 | 0.9904 | 0.9899 ± 0.0028 | **0.9934 ± 0.0032** |
| | 25 | 25 | 36 | 0.9983 ± 0.0006 | 0.9989 | 0.9987 | 0.9982 ± 0.0006 | **0.9992 ± 0.0003** |
| | 50 | 50 | 36 | 0.9997 ± 0.0001 | 0.9999 | 0.9998 | 0.9996 ± 0.0001 | **0.9999 ± 0.0000** |
| | 113 | 28 | 36 | 1.0000 ± 0.0000 | 1.0000 | **1.0000** | 0.9999 ± 0.0000 | 1.0000 ± 0.0000 |
| Bank | 50 | 50 | 9043 | 0.6762 ± 0.0000 | 0.7096 | 0.6673 | **0.6942 ± 0.0000** | 0.6762 ± 0.0000 |
| | 100 | 100 | 9043 | 0.6651 ± 0.0273 | 0.7140 | 0.6344 | **0.6917 ± 0.0160** | 0.6884 ± 0.0306 |
| | 250 | 250 | 9043 | 0.6908 ± 0.0164 | 0.7585 | 0.6899 | **0.7202 ± 0.0242** | 0.7180 ± 0.0210 |
| | 500 | 500 | 9043 | 0.7305 ± 0.0131 | 0.7834 | 0.7236 | 0.7662 ± 0.0091 | **0.7730 ± 0.0086** |
| | 28934 | 7233 | 9043 | 0.7820 ± 0.0015 | 0.7884 | 0.7820 | **0.7925 ± 0.0019** | 0.7861 ± 0.0022 |
| Blood | 10 | 10 | 150 | 0.5355 ± 0.0144 | 0.5578 | 0.5329 | 0.5465 ± 0.0086 | **0.5592 ± 0.0115** |
| | 25 | 25 | 150 | 0.5208 ± 0.0279 | 0.5397 | 0.5163 | 0.5217 ± 0.0227 | **0.5278 ± 0.0299** |
| | 50 | 50 | 150 | 0.5299 ± 0.0213 | 0.5447 | 0.5307 | 0.5238 ± 0.0145 | **0.5491 ± 0.0140** |
| | 100 | 100 | 150 | 0.5231 ± 0.0155 | 0.5424 | 0.5311 | 0.5342 ± 0.0137 | **0.5404 ± 0.0108** |
| | 250 | 250 | 150 | 0.5400 ± 0.0091 | 0.5412 | 0.5393 | **0.5498 ± 0.0050** | 0.5416 ± 0.0077 |
| | 478 | 119 | 150 | 0.5367 ± 0.0049 | 0.5497 | 0.5350 | **0.5395 ± 0.0032** | 0.5384 ± 0.0063 |
| CalHousing | 10 | 10 | 4128 | 0.7472 ± 0.0334 | 0.7972 | 0.7821 | 0.7521 ± 0.0191 | **0.7998 ± 0.0112** |
| | 25 | 25 | 4128 | 0.8260 ± 0.0063 | 0.8744 | 0.8242 | 0.8576 ± 0.0064 | **0.8731 ± 0.0081** |
| | 50 | 50 | 4128 | 0.8335 ± 0.0168 | 0.9002 | 0.8311 | 0.8799 ± 0.0098 | **0.8938 ± 0.0085** |
| | 100 | 100 | 4128 | 0.8579 ± 0.0050 | 0.9179 | 0.8590 | 0.9002 ± 0.0048 | **0.9147 ± 0.0045** |
| | 250 | 250 | 4128 | 0.8836 ± 0.0046 | 0.9302 | 0.8826 | 0.9241 ± 0.0030 | **0.9277 ± 0.0029** |
| | 500 | 500 | 4128 | 0.8977 ± 0.0067 | 0.9329 | 0.8967 | **0.9296 ± 0.0028** | 0.9283 ± 0.0028 |
| | 13209 | 3302 | 4128 | 0.9179 ± 0.0018 | 0.9374 | 0.9137 | **0.9459 ± 0.0009** | 0.9365 ± 0.0017 |
| Credit-g | 10 | 10 | 200 | 0.5729 ± 0.0368 | 0.5981 | 0.5817 | 0.5754 ± 0.0383 | **0.5849 ± 0.0312** |
| | 25 | 25 | 200 | 0.6382 ± 0.0146 | 0.6328 | **0.6428** | 0.6408 ± 0.0136 | 0.6399 ± 0.0086 |
| | 50 | 50 | 200 | 0.6687 ± 0.0164 | 0.6349 | 0.6649 | 0.6490 ± 0.0103 | **0.6659 ± 0.0152** |
| | 100 | 100 | 200 | 0.7099 ± 0.0089 | 0.6721 | 0.7094 | 0.6860 ± 0.0070 | **0.7122 ± 0.0102** |
| | 250 | 250 | 200 | 0.7442 ± 0.0084 | 0.7151 | **0.7504** | 0.7181 ± 0.0047 | 0.7489 ± 0.0066 |
| | 640 | 160 | 200 | 0.7763 ± 0.0016 | 0.7485 | 0.7770 | 0.7444 ± 0.0026 | **0.7845 ± 0.0011** |
| Diabetes | 10 | 10 | 154 | 0.6756 ± 0.0513 | 0.6971 | 0.6794 | 0.6806 ± 0.0508 | **0.6890 ± 0.0558** |
| | 25 | 25 | 154 | 0.7582 ± 0.0160 | 0.7778 | 0.7699 | **0.7814 ± 0.0168** | 0.7728 ± 0.0218 |
| | 50 | 50 | 154 | 0.7895 ± 0.0082 | 0.8196 | 0.7879 | **0.8091 ± 0.0062** | 0.8004 ± 0.0122 |
| | 100 | 100 | 154 | 0.8091 ± 0.0041 | 0.8357 | 0.8103 | **0.8254 ± 0.0048** | 0.8182 ± 0.0077 |
| | 250 | 250 | 154 | 0.8244 ± 0.0033 | 0.8431 | 0.8203 | 0.8365 ± 0.0034 | **0.8396 ± 0.0037** |
| | 491 | 122 | 154 | 0.8369 ± 0.0034 | 0.8530 | 0.8376 | **0.8483 ± 0.0008** | 0.8383 ± 0.0035 |
| Heart | 10 | 10 | 184 | 0.7289 ± 0.0667 | 0.8206 | 0.8073 | 0.7096 ± 0.0705 | **0.8141 ± 0.0148** |
| | 25 | 25 | 184 | 0.8150 ± 0.0182 | 0.8372 | 0.8214 | 0.8284 ± 0.0083 | **0.8334 ± 0.0117** |
| | 50 | 50 | 184 | 0.8406 ± 0.0078 | 0.8455 | 0.8405 | 0.8400 ± 0.0089 | **0.8485 ± 0.0075** |
| | 100 | 100 | 184 | 0.8490 ± 0.0052 | 0.8470 | 0.8470 | 0.8495 ± 0.0023 | **0.8512 ± 0.0044** |
| | 250 | 250 | 184 | 0.8694 ± 0.0032 | 0.8633 | 0.8712 | 0.8680 ± 0.0046 | **0.8734 ± 0.0041** |
| | 587 | 146 | 184 | 0.8749 ± 0.0013 | 0.8721 | **0.8777** | 0.8734 ± 0.0013 | 0.8766 ± 0.0006 |
| Jungle | 10 | 10 | 8964 | 0.6638 ± 0.0141 | 0.6938 | 0.6688 | **0.6761 ± 0.0130** | 0.6683 ± 0.0170 |
| | 25 | 25 | 8964 | 0.7098 ± 0.0130 | 0.7275 | 0.7176 | **0.7257 ± 0.0137** | 0.7169 ± 0.0138 |
| | 50 | 50 | 8964 | 0.7613 ± 0.0086 | 0.7536 | 0.7590 | 0.7577 ± 0.0065 | **0.7649 ± 0.0086** |
| | 100 | 100 | 8964 | 0.8014 ± 0.0045 | 0.7894 | 0.8038 | 0.7941 ± 0.0035 | **0.8056 ± 0.0050** |
| | 250 | 250 | 8964 | 0.8271 ± 0.0061 | 0.8151 | 0.8231 | 0.8138 ± 0.0067 | **0.8336 ± 0.0062** |
| | 500 | 500 | 8964 | 0.8431 ± 0.0062 | 0.8352 | 0.8372 | 0.8354 ± 0.0037 | **0.8528 ± 0.0057** |
| | 28684 | 7171 | 8964 | 0.9096 ± 0.0024 | 0.8425 | 0.9078 | 0.8877 ± 0.0022 | **0.9094 ± 0.0023** |



## C SUMMARISED RESULTS

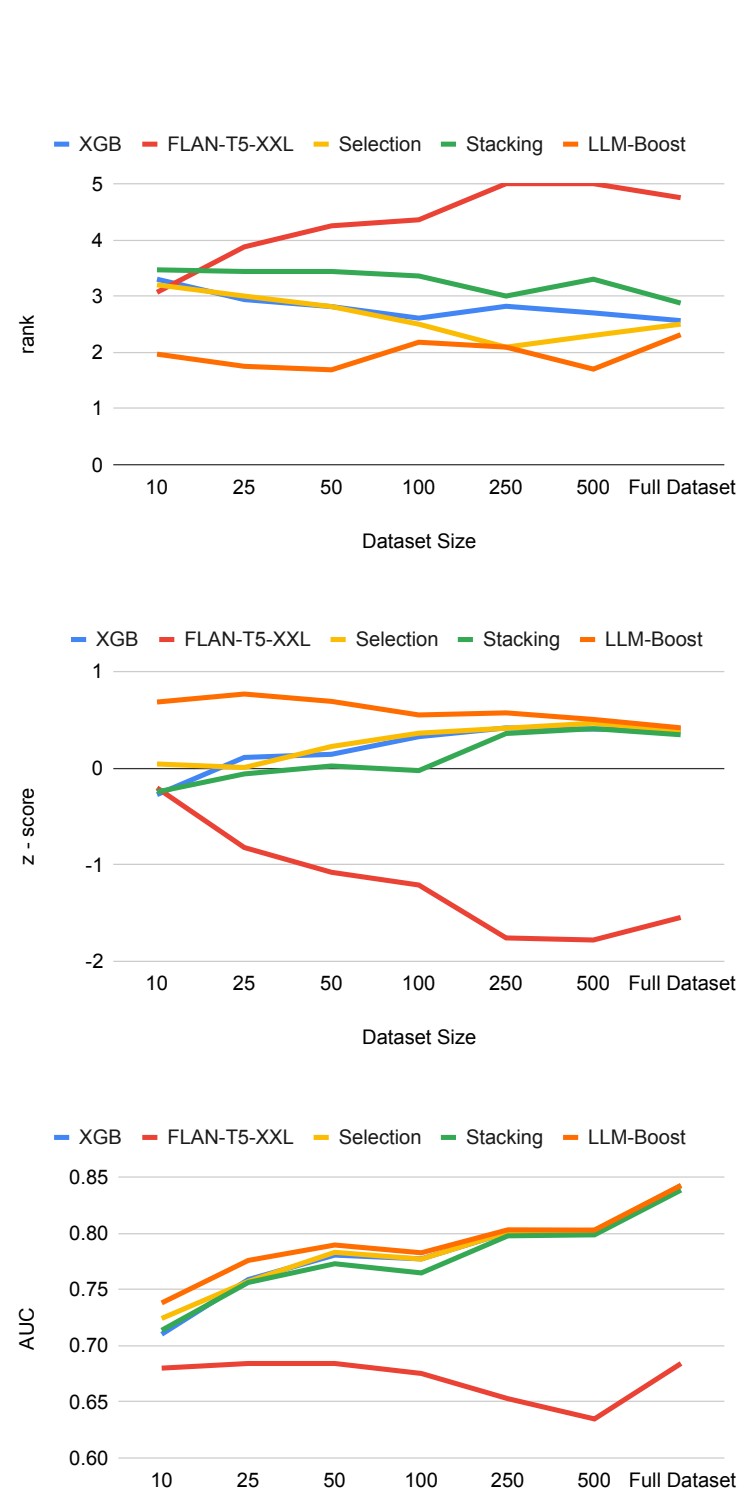

Figure 7: Average AUC based statistical metrics of XGBoost + Flan-T5-XXL

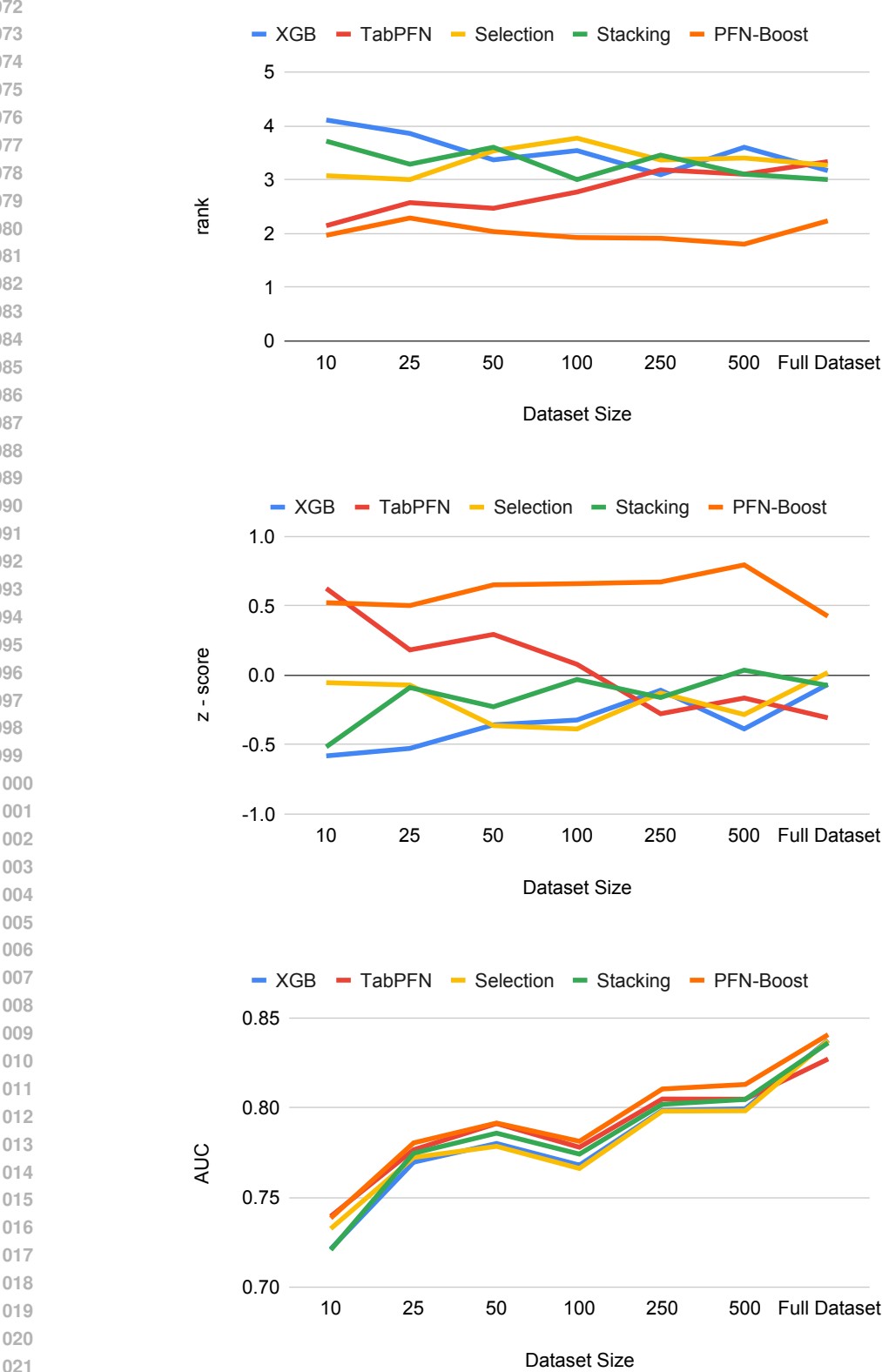

Figure 8: Average AUC based statistical metrics of XGBoost + TabPFN

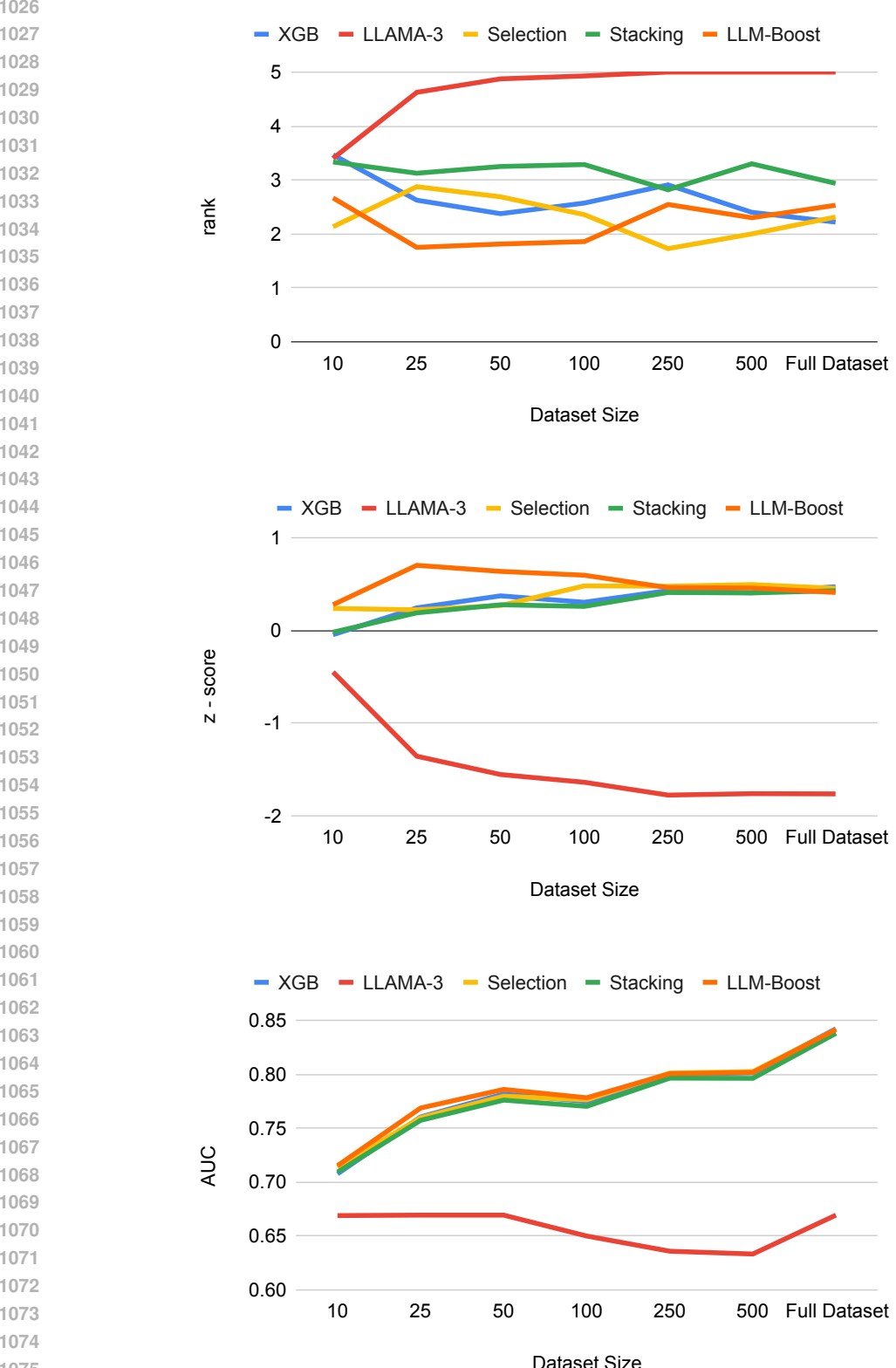

Figure 9: Average AUC based statistical metrics of XGBoost + Meta-Llama-8B-Instruct

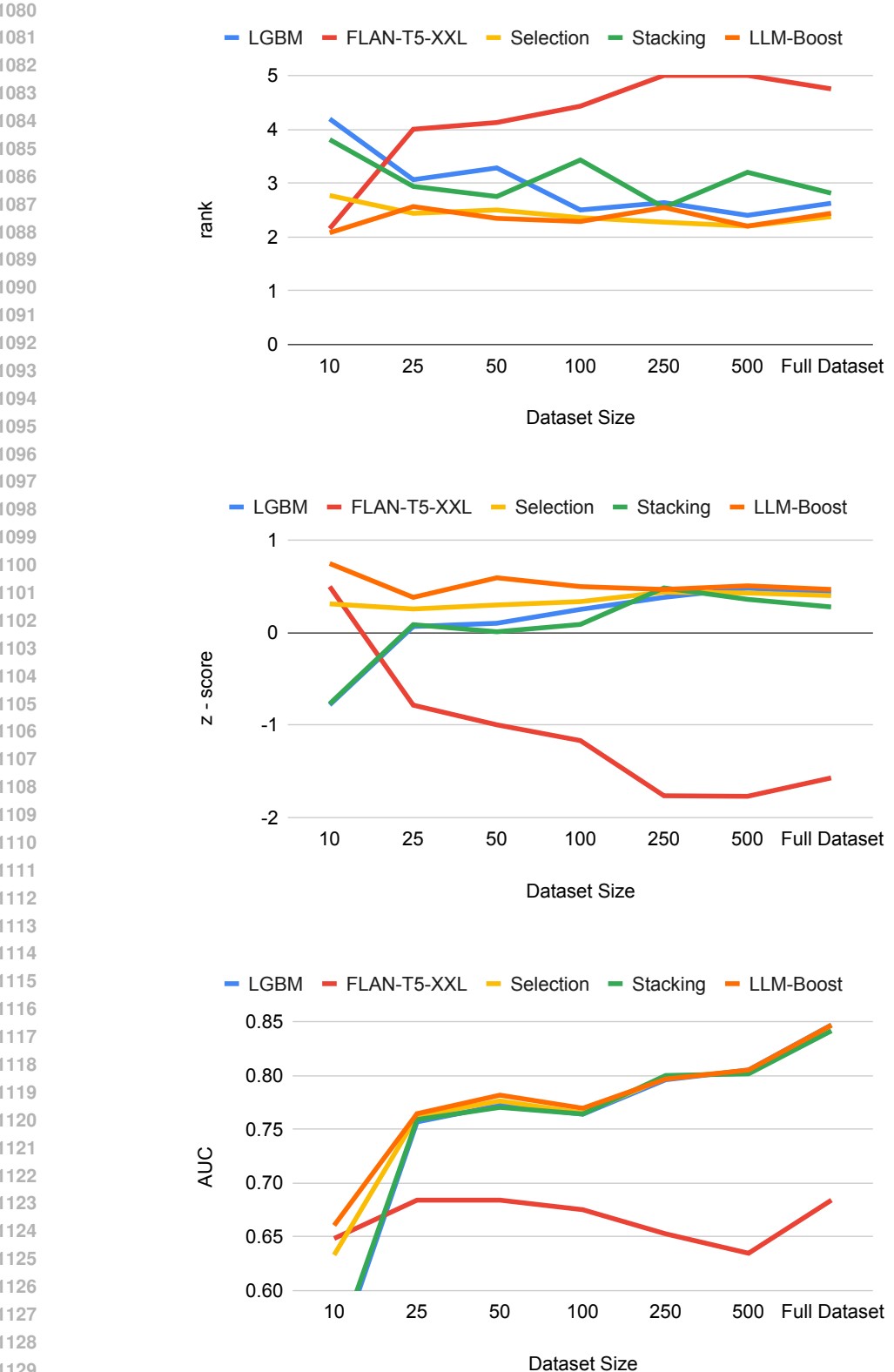

Figure 10: Average AUC based statistical metrics of LightGBM + Flan-T5-XXL

# D   COMPARISON WITH TABLLM

| Dataset | Size | LLM-Boost | Tabllm |
|---|---|---|---|
| bank | 4 | 0.57 | **0.59** |
| | 8 | 0.61 | **0.64** |
| | 16 | 0.62 | **0.65** |
| | 32 | **0.76** | 0.64 |
| | 64 | **0.79** | 0.69 |
| | 128 | **0.83** | 0.82 |
| | 256 | 0.87 | 0.87 |
| | 512 | 0.88 | 0.88 |
| blood | 4 | 0.54 | **0.58** |
| | 8 | 0.58 | **0.66** |
| | 16 | 0.58 | **0.66** |
| | 32 | 0.61 | **0.68** |
| | 64 | 0.66 | **0.68** |
| | 128 | 0.67 | **0.68** |
| | 256 | 0.68 | **0.70** |
| | 512 | 0.68 | 0.68 |
| calhousing | 4 | 0.62 | **0.63** |
| | 8 | **0.69** | 0.60 |
| | 16 | **0.75** | 0.70 |
| | 32 | **0.78** | 0.77 |
| | 64 | **0.80** | 0.77 |
| | 128 | **0.84** | 0.81 |
| | 256 | **0.86** | 0.83 |
| | 512 | **0.90** | 0.86 |
| creditg | 4 | 0.61 | **0.69** |
| | 8 | **0.69** | 0.66 |
| | 16 | 0.65 | **0.66** |
| | 32 | 0.68 | **0.72** |
| | 64 | **0.71** | 0.70 |
| | 128 | **0.72** | 0.71 |
| | 256 | **0.73** | 0.72 |
| | 512 | **0.76** | 0.70 |
| diabetes | 4 | 0.58 | **0.61** |
| | 8 | 0.63 | 0.63 |
| | 16 | 0.67 | **0.69** |
| | 32 | **0.74** | 0.68 |
| | 64 | **0.77** | 0.73 |
| | 128 | **0.80** | 0.79 |
| | 256 | **0.80** | 0.78 |
| | 512 | **0.80** | 0.78 |
| heart | 4 | 0.64 | **0.76** |
| | 8 | 0.77 | **0.83** |
| | 16 | 0.82 | **0.87** |
| | 32 | 0.88 | 0.87 |
| | 64 | 0.90 | **0.91** |
| | 128 | **0.91** | 0.90 |
| | 256 | 0.92 | 0.92 |
| | 512 | 0.92 | 0.92 |
| income | 4 | 0.63 | **0.84** |
| | 8 | 0.75 | **0.84** |
| | 16 | 0.79 | **0.84** |
| | 32 | 0.84 | 0.84 |
| | 64 | 0.82 | **0.84** |
| | 128 | 0.87 | **0.86** |
| | 256 | 0.87 | 0.87 |
| | 512 | 0.88 | **0.89** |
| jungle | 4 | 0.61 | **0.64** |
| | 8 | 0.62 | **0.64** |
| | 16 | 0.64 | **0.65** |
| | 32 | **0.74** | 0.71 |
| | 64 | 0.75 | **0.78** |
| | 128 | **0.82** | 0.81 |
| | 256 | **0.85** | 0.84 |
| | 512 | 0.88 | **0.89** |

# E    EXAMPLE ILLUSTRATING THE IMPORTANCE OF THE SCALING PARAMETER FOR LLM-BOOST

For LLM-Boost the predictions of the ensemble consisting of the first $i$ trees are,

$$pred_{(0,i)} = pred_{(1,i)} + s * \text{SCORE}_{\text{LLM}} + C,$$

The following graph highlights the importance of tuning the scaling parameter $s$ for our boosting framework.

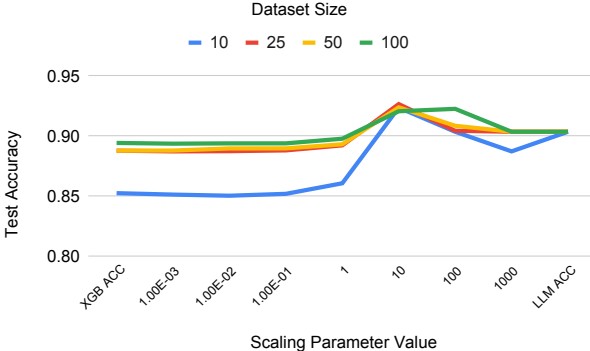

Figure 11: **LLM-Boost with intermediate scaling parameter values may lead to better perfor­mance than either standalone model.** This ablation study demonstrates the change in performance with the scaling parameter when boosting a pre-fine tuned XGBoost model with Flan-T5 scores. The x-axis represents the scaling parameter which ranges from 0 to infinity. When the scaling parameter is close to 0, the performance approaches that of XGBoost, since the seed values in LLM-Boost are negligent. As the scaling parameter increases, we approach the raw performance of the LLM. We observe how with LLM-Boost, intermediate scaling values result in better performance than either the individual LLM or GBDT algorithm.

