# OpenReview forum: "LLMs Boost the Performance of Decision Trees on Tabular Data across Sample Sizes"
_ICLR.cc/2025/Conference — Submitted to ICLR 2025_

### Official Review · Reviewer_h4aV · 2024-10-27

**Soundness:** 3
**Presentation:** 2
**Contribution:** 2
**Rating:** 3
**Confidence:** 5

**Summary:**

The authors propose LLM-Boost, a fusion of LLM and decision tree algorithms. The main motivation here is the idea that GBDT is known to be the best performing predictive model in the tabular domain, but it cannot handle linguistic context, so it would be useful to be able to inject linguistic context into GBDT.

**Strengths:**

1. Great motivation. I also agree that injecting linguistic context is important to improve tabular predictive models, and to that end, many researchers are trying to incorporate LLM, which is an important research direction in the tabular learning community.

2. Simple method. If I understood the paper clearly, the main idea of LLM-Boost is to change the first step of GBDT determination to LLM's prediction, which seems really simple and intuitive to me.

3. The authors conducted extensive experiments on a variety of datasets.

**Weaknesses:**

1. Visualizations for the main results are too difficult to understand.

2. In some cases, the performance improvement is too small or even no improvement. While the authors claim that it is important to integrate LLM and GBDT as the dataset size increases, results such as Table 3 (Appendix) do not support this. When the training set size is at its maximum, the performance of LLM-Boost is almost identical to XGBoost.

**Questions:**

1. Did the authors try other LLMs, such as Llama 3?

2. Why did the authors use Flan-T5 for the main experiment?

3. Can the proposed LLM-Boost be used with commercial-level LLMs like GPT-4 (which is a black-box API)?

4. Have you tried a dataset with more samples? For example, a dataset with a million samples.

---

> ### Author Response · Authors · 2024-11-17
> **Thank you for your review**
>
> Thank you for identifying the simplicity of our approach and your insightful review. We hope our response below fully addresses your concerns and questions. Please let us know if there is anything else we can do to make our submission stronger or if you have any additional questions.
>
> W1. We understand the average rank and z-score metrics are somewhat derivative but these metrics are more robust than directly comparing AUC (which we also provide) due to reasons we have highlighted in the first few paragraphs of section 5. For example, some datasets have higher variance over AUC than others and those would dominate average AUC numbers. Additionally, average rank is commonly used to compare performance in the tabular domain. If you can elaborate further we will include your suggestions to improve the interpretability of our visualizations.
>
> W2. We thank the reviewer for this comment. LLM-Boost makes most sense when the dataset size is too large for the LLMs (due to context length limits) and two small for the GBDTs, so each standalone model can still benefit from the other. We demonstrate this through our experiments for intermediate dataset sizes. We further note that LLM-Boost does no worse than the GBDT even when the dataset size is large. We have changed the writing in the introduction and abstract of our revised submission to make this point clearer.
>
> Q1&2.  Given our resource constraints, the 11 billion parameter Flan-T5-XXL model was the largest open source LLM we were able to run inference on. We also experimented with Llama-3-8B-Instruct, but we found that this model was less robust to the prompt and performance was worse in general. We believe that Flan-T5-XXL already demonstrates the benefits of our method against baselines. Detailed experimental results for both these LLMs are provided in our Appendix.
>
> Q3. For our method, we need the loss/log-probabilities to calculate the LLM scores. The current GPT-4 API provides log-probabilities for a limited number of most likely tokens at each position. This might be sufficient for some prompts but some additional engineering effort (perhaps using multiple partial API calls along with logit bias) might be required to address corner cases as we need the log probabilities for all classes.
>
> Q4. We have not tried very large datasets as we already observe the convergence of our method to the GBDT performance at the largest dataset scales we examine.  The core strength of our method is in the intermediate dataset sizes.
>
> We once again thank you for your feedback. We hope we have adequately addressed your concerns and questions. We would appreciate it if you would consider increasing your score in light of our response.

---

> > ### Comment · Reviewer_h4aV · 2024-11-22
> > **Thank you for your response**
> >
> > I appreciate the time and effort you've put into addressing my concerns. However, my concerns have not been fully addressed.
> >
> > - First of all, the reason I said that the visualization is hard to understand is because the difference between each baseline and your method is not clearly distinguished, which I think is due to the minimal improvement in LLM-Boost.
> >
> > - Second, I think the authors need to clearly define the scope: as I recall, in the first draft, it seemed to cover the full range of tabular datasets, but now the authors claim that their main contribution is on intermediate-sized datasets - how do they define “intermediate-sized” and is this really a practical situation?
> >
> > - Finally, I also think that Llama-3-8B is a better modern model than Flat-T5-XXL. The authors' observation is that Llama-3-8B is generally worse to run LLM-Boost on, but is this in a way that is incompatible across different LLMs? If so, once a better open LLM is released, LLM-Boost will not be able to benefit from it.

---

> > > ### Author Response · Authors · 2024-11-26
> > >
> > > We really appreciate your response and willingness to engage with us.
> > >
> > > * We believe that our algorithm clearly outperforms the other baselines in our main results in terms of AUC, Rank, and Z-Score for the vast majority of tested dataset sizes, sometimes by a large margin. (Figure 3 and Figure 4)
> > > We actually only added the word “intermediate” to the abstract on this draft and did not change this anywhere else. In the case of boosted LLMs, LLM-Boost performance converges to the GBDT performance as datasets grow beyond 1000 samples.  Due to the significant performance gap between the few-shot LLM and the GBDT, we quickly learn scaling hyperparameters $s$ which increasingly fall back on the GBDT.  However, we observe this convergence to GBDT performance happens much later in the case of boosted TabPFN as it performs much better than standalone GBDT in many cases even at large dataset sizes. To remove the vagueness of “intermediate-sized” we have again slightly revised the abstract to state “While matching LLMs at sufficiently small dataset sizes and GBDTs at sufficiently large sizes, LLM-Boost outperforms both standalone models on a wide range of dataset sizes in between”.
> > > * We found that Llama-3-8B-Instruct was often unable to follow our instructions to output only the class label, making prompt engineering more difficult than for Flan-T5-XXL. This instruction is not fundamentally difficult to follow as we simply ask the model to only write the class label. We found that this particular version of Llama tends to prefer longer responses. Flan-T5 is tuned better for these types of simple instructions and short responses, but in principle any high-quality language model could also be fine-tuned to output short completions. Another strategy could be to parse the text output by Llama-3-8B-Instruct and extract the answer.  We should also note that the Flan-T5-XXL is a larger model with more parameters than Llama-3-8B-Instruct, although you are absolutely correct that it is an older and more primitive model.  On the LiveBench leaderboard, small open-weights LLMs including Llama-3-8B-Instruct score very poorly at instruction following compared to large models like Llama-3.1-70b-instruct-turbo, which is unaffordable for us on our tiny compute budget, but other LLMs with better instruction following may outperform Flan-T5. Finally, we stress that even though Flan-T5-XXL is an older model, our method nonetheless outperforms existing alternatives, like XGBoost, which are in wide use across industries. Therefore, we feel that our method is already highly useful and would be valuable for the ICLR community.
> > >
> > > We once again thank you for your time and feedback. Please reach-out again if there are any additional questions.

---

### Official Review · Reviewer_Sk2m · 2024-10-27

**Soundness:** 1
**Presentation:** 2
**Contribution:** 1
**Rating:** 3
**Confidence:** 4

**Summary:**

The paper introduces LLM-Boost, a method that combines large language models (LLMs) and gradient-boosted decision trees (GBDTs) to improve performance on tabular data. The approach leverages LLMs' ability to interpret column headers and GBDTs' efficiency on large datasets. LLMs are used to extract predictions and residuals, which are then refined by GBDT models, leading to improvements, particularly on small to medium-sized datasets.

**Strengths:**

1. **Novelty.** Demonstrating that combining LLMs with GBDTs for tabular prediction tasks in such a simple manner can be effective is a valuable contribution.

2. **Discussion on limitations and future work.** A thorough discussion of limitations and future work is often overlooked, but it is something researchers reading the paper can greatly appreciate.

**Weaknesses:**

1. **Computational overhead.** It is unclear how practical it is to use LLM inference for tabular prediction tasks. An alternative use of LLMs in the tabular domain, which arguably is much simpler conceptually, is to use them for automatic feature engineering and training GBDTs on the augmented set of features for prediction [1, 2].

2. **Missing experiments.** Several crucial baselines, such as comparing LLM-Boost against simply increasing the number of trees in the GBDT, are missing. Additionally, it would be useful to evaluate how the method scales to datasets with significantly larger feature sets.

---
[1] Hollmann et al. “Large language models for automated data science: Introducing caafe for context-aware automated feature engineering.” (2023).\
[2] Nam et al. “Optimized Feature Generation for Tabular Data via LLMs with Decision Tree Reasoning.” (2024).

**Questions:**

1. How does LLM-Boost perform when column headers are semantically ambiguous or incorrect? Would it still provide performance benefits over standalone GBDTs?

2. Why can’t we simply compare prediction accuracies to evaluate performance?

3. Typos: In Section 3.2, “replacing the first tree int he” → “replacing the first tree in the”.

---

> ### Author Response · Authors · 2024-11-17
> **Thank you for your review**
>
> Thank you for identifying the novelty of our approach and your insightful review. We hope our response below fully addresses your concerns and questions. Please let us know if there is anything else we can do to make our submission stronger or if you have any additional questions.
>
> W1. We agree with the reviewer that automatic feature engineering methods using LLMs are another promising approach. However, we believe that these methods are orthogonal to each other and can be used in a complementary fashion.  Future work may consider using LLMs both to engineer new features and then make predictions on those engineered features with LLM-based predictors.
>
> W2. We thank the reviewer for these thoughtful comments. We consider the number of trees in the GBDT as a hyper-parameter which is optimized during HPO for the LLM-Boost and standalone GBDT results. Additionally, we believe that our method is not really suitable for datasets with considerably larger feature set sizes as the LLM context length limit will be a bottleneck.
>
> Q1. Thank you for noting this observation, we actually tested out the dependence of LLM-Boost on interpretable column headers in Section 5.4 and have demonstrated that LLM-Boost still outperforms the standalone GBDT even when the column headers are not meaningful. The LLM is still free to infer the meaning of columns from the data values themselves, especially for categorical features.
>
> Q2. We choose AUC instead of accuracy as it is a widely adopted metric in the tabular domain and is better for datasets with class imbalance. While we report average AUC, we additionally include z-score and rank as we believe they are more robust indicators of performance due to reasons highlighted in the first few paragraphs of Section 5.  For example, some datasets yield higher variance of accuracies than others and those would dominate average accuracy numbers.
>
> Q3. Thank you, we have added these corrections to our revised submission.
>
> We once again thank you for your feedback. We hope we have adequately addressed your concerns and questions. We would appreciate it if you would consider increasing your score in light of our response.

---

> > ### Comment · Reviewer_Sk2m · 2024-11-23
> >
> > Thank you for your response.
> >
> > [W1, W2] While it is true that LLMs can be used both for offline feature engineering and online predictions, avoiding the latter by relying solely on the former would be significantly more practical, as it eliminates the potentially high costs of LLM inference. As noted in Section 3, the method can be interpreted as "replacing the first tree of the GBDT ensemble with the static prediction of the LLM," which suggests that what LLM is essentially providing in LLM-Boost is providing additional features that enhance the GBDT's predictions. This leads me to believe that it is indeed feasible to avoid online LLM inference by leveraging LLMs for offline feature engineering. While this may not turn out to be the case, a direct comparison would provide stronger evidence of LLM-Boost’s practical value.

---

> > > ### Author Response · Authors · 2024-11-27
> > >
> > > Thank you very much for further engaging with us.
> > >
> > > While both approaches leverage LLMs to improve tabular predictions, there are critical differences between our approaches, each having advantages and disadvantages.  LLM feature engineering may only require running a GBDT at inference, which is an advantage for the subset of tabular tasks where inference latency is critical.  On the other hand, the LLM usually only uses a tiny fraction of the table’s samples and can only pass a small amount of information to the GBDT (the formulas for new features, which may have a hard time capturing rich textual information from column headers).  In our boosting method, the LLM uses all training data, can pass rich information to the GBDT, and can combine strong models with GBDTs other than LLMs, potentially leading to much stronger performance yet slower inference latency.  We do point out that practitioners use LLMs for inference on a wide range of tasks these days so whether or not those inference costs are a problem will vary by application.
> > >
> > > We agree with the reviewer that a direct comparison of our method with feature engineering will be a compelling addition to our paper.  We are currently implementing this comparison and will include it in the camera-ready deadline.  Note that [1] conducts their main results with GPT-4 and uses different data splits than we do, and [2] does not provide code, so we are reimplementing their method and making sure all experiments are fair (on the same train-test splits, etc.).
> > >
> > > Please let us know if you have any more questions.
> > >
> > > [1] Hollmann et al. “Large language models for automated data science: Introducing caafe for context-aware automated feature engineering.” (2023).
> > >
> > > [2] Nam et al. “Optimized Feature Generation for Tabular Data via LLMs with Decision Tree Reasoning.” (2024).

---

> ### Author Response · Authors · 2024-11-27
>
> In addition, inspired by your feedback we have added the following paragraph on LLM-based feature engineering approaches to our related works section.
>
> "Alternatively, LLMs have been used for automatic feature engineering in the tabular domain. Lightweight models, such as GBDTs, that are then trained on the augmented set of features have demonstrated superior performance to those trained on the original features Hollmann et al. (2023b); Nam et al. (2024). While this approach is computationally efficient at inference-time compared to our proposed procedure which uses the LLM during inference, the LLM typically only utilizes a small fraction of the table’s samples to generate new features. Additionally, this approach usually requires powerful API models to be effective Hollmann et al. (2023b)."
>
> We appreciate your feedback and believe that it has made our submission stronger. If our response adequately addressed your concerns please consider increasing your score.

---

### Official Review · Reviewer_Fo2b · 2024-11-03

**Soundness:** 2
**Presentation:** 2
**Contribution:** 1
**Rating:** 3
**Confidence:** 4

**Summary:**

This paper combines both LLM and decision trees to do the classification task on the tabular data. The major experiment is to explore how LLM, decision tree and the combined method work across different data sizes.

**Strengths:**

1. This paper combines LLM and decision trees for tabular classification tasks. It shows certain novelty in the proposed method.

**Weaknesses:**

1. The major baseline is quite strange. It would be better to separately show how LLM and decision tree work and treat both as the baselines, rather than mix them into one baseline 'Select'.
2. The experiment is not comprehensive. This paper simply lists the performance of different data sizes. It would be better to show some error analysis and cases to indicate when the proposed method can excel.
3. The selection of the LLM (eg, Flan-T5) is not very convincing.

**Questions:**

1. Can close-sourced LLMs such as GPT-family models perform well on such tasks?
2. What is the rationale for choosing Flan-T5 for the major experiments?

---

> ### Author Response · Authors · 2024-11-17
> **Thank you for your review**
>
> Thank you for identifying the novelty of our method. We believe we have improved the interpretability of our submission with your feedback. We hope our response below fully addresses your concerns and questions. Please let us know if there is anything else we can do to make our submission stronger or if you have any additional concerns.
>
> W1. We thank the reviewer for this suggestion. We actually did include the constituent standalone models in our AUC results. (See the right subfigures of figures 3 and 4). As the standalone models performed worse compared to the ensemble baselines especially for the boosted LLM results we omitted them from our z-score and rank comparisons. Nonetheless, we see your point and have added them back to all our results in our revised submission.
>
> W2. We have demonstrated that LLM-Boost provides a considerable boost in performance on intermediate dataset sizes where both the standalone few-shot LLM and GBDT have limitations. This is the reason we have chosen to present average metrics for each dataset size in our main result. If you can further elaborate on any specific experiments you might have in mind, we are happy to add them and improve interpretability of our results.
>
> W3. Given our resource constraints, the 11 Billion parameter Flan-T5-XXL model was the largest open source LLM we were able to run inference on. We also experimented with Llama-3-8B-Instruct but we found prompt engineering was more challenging for this model and performance was worse in general. However, we still believe that Flan-T5-XXL already demonstrates the benefits of our method against baselines.
>
> Q1. For our method, we need the loss/log-probabilities to calculate the LLM scores. The current GPT-4 API provides log-probabilities for a limited number of most likely tokens at each position. This might be sufficient for some prompts but some additional engineering effort (perhaps using multiple partial API calls along with logit bias) might be required to address corner cases as we need the log probabilities for all classes.
>
> Q2. The computational cost for very-large LLM inference overshadows that required for GBDT training. With our resource constraints in mind, we decided on LLMs with around 10 billion parameters, which fit on a single GPU. We conducted experiments with the Flan-T5-XXL and Llama-3-8B-Instruct model but found that the Flan-T5 model performed better. Detailed experimental results for both these LLMs are provided in our Appendix.
>
> We once again thank you for your feedback. We hope we have adequately addressed your concerns and questions. We would appreciate it if you would consider increasing your score in light of our response.

---

### Official Review · Reviewer_2C18 · 2024-11-10

**Soundness:** 3
**Presentation:** 3
**Contribution:** 3
**Rating:** 5
**Confidence:** 4

**Summary:**

The paper introduces the LLM-Boost algorithm, which integrates large language models (LLMs) with gradient-boosted decision trees (GBDTs) to enhance classification performance on tabular datasets. The method involves extracting LLM logits for each row of data and using these predictions to augment the GBDT model, allowing it to learn the residuals to the true labels. The authors demonstrate that LLM-Boost outperforms traditional models and other ensemble techniques across various dataset sizes, showcasing its potential in automating predictive modeling pipelines.

**Strengths:**

- **Enhanced Performance:** LLM-Boost shows superior classification performance compared to traditional GBDTs and other ensemble methods.
- **Model Agnostic:** The boosting mechanism can be applied to various high-performing tabular architectures beyond LLMs.
- **Efficiency in Training:** Once LLM outputs are pre-computed, the training cost aligns with that of standard GBDT training.

**Weaknesses:**

- **Dependence on Interpretability:** The method requires interpretable text descriptors as column headers, which may necessitate prompt engineering for some datasets.
- **Pre-computation Costs:** For very large datasets, the initial cost of pre-computing LLM outputs may become significant.

**Questions:**

1. What is the performance with large-sized datasets, instead of small and medium sized datasets?
2. line 201 "weather" -> "whether"
3. Missing inference:
- Iida, Hiroshi, Dung Thai, Varun Manjunatha, and Mohit Iyyer. "Tabbie: Pretrained representations of tabular data." arXiv preprint arXiv:2105.02584 (2021).
- Chen, Pei, Soumajyoti Sarkar, Leonard Lausen, Balasubramaniam Srinivasan, Sheng Zha, Ruihong Huang, and George Karypis. "HYTREL: Hypergraph-enhanced tabular data representation learning." Advances in Neural Information Processing Systems 36 (2024).

---

> ### Author Response · Authors · 2024-11-17
> **Thank you for your review.**
>
> Thank you for the encouraging review and feedback. We believe we have improved our submission with your suggestions. We hope our response below fully addresses your concerns and questions. Please let us know if there is anything else we can do to make our submission stronger or if you have any additional concerns.
>
> W1. This is a keen observation. We actually tested out the dependence of LLM-Boost on interpretable column headers in Section 5.4, and demonstrated that LLM-Boost still outperforms the standalone GBDT even when the column headers are not meaningful. The LLM is still free to infer the meaning of columns from the data values themselves, especially for categorical features.
>
> W2. Thank you for pointing this out, we believe that LLM-Boost is best suited for dataset sizes which are too big for few-shot LLMs and too small for GBDTs to perform well as we have mentioned in our introduction (line 58). We have also slightly amended our abstract to highlight this fact.
>
> Q1. We have demonstrated that LLM-Boost does not perform any worse than the standalone GBDT even for large datasets where the GBDTs massively outperform the few-shot LLM. However, we again re-iterate that the LLM-Boost provides the most benefit at intermediate dataset sizes where both standalone components perform well. Otherwise, one component will have little to offer.
>
> Q2&3. Thank you for pointing out these corrections. We have revised our submission.
>
> We once again thank you for your feedback. We hope we have adequately addressed your concerns and questions. We would appreciate it if you would consider increasing your score in light of our response.

---

> > ### Comment · Reviewer_2C18 · 2024-12-01
> >
> > Thank you for the response. I will keep my rating.

---

### Meta-Review · Area_Chair_sQji · 2024-12-23

**Metareview:**

This paper presents LLM-Boost, an algorithm that integrates large language models (LLMs) with gradient-boosted decision trees (GBDTs) for classification tasks on tabular data. The proposed method replaces the first tree of the GBDT ensemble with the static prediction of the LLM. The authors argue that this hybrid method can leverage the strengths of both LLMs (textual/context understanding) and GBDTs (robust performance on tabular data, especially with sufficient samples). Experiments compare the proposed method against baseline models and ensemble techniques across various dataset sizes, with the conclusion that LLM-Boost excels particularly in small to medium-sized datasets.

**Strengths:**

* "Simple and intuitive" method (h4aV), with "great motivation" (h4aV) and "valuable contribution" (Sk2m) to use a hybrid approach to combine LLMs and classical GDBT methods.

* The proposed method shows potential of "superior classification performance compared to traditional GBDTs and other ensemble methods" (2C18), while the improvements largely come from intermediate-sized datasets (see weaknesses for details), leading to concerns around its limited practical impact.

* The authors explicitly discussed the limitations of the proposed approach, which is "often overlooked, but it is something researchers reading the paper can greatly appreciate." (Sk2m).

* "extensive experiments on a variety of datasets." – (h4aV)

**Weaknesses:**

There are two major issues of this paper:

* First of all, it is unclear whether the focus on "intermediate" dataset sizes would limit the practical impact of the paper. As noted by h4aV:

> The authors need to clearly define the scope. … now the authors claim that their main contribution is on intermediate-sized datasets – how do they define ‘intermediate-sized’ and is this really a practical situation?

while I believe this setting is important, the authors need to provide further clarification along with analysis to clearly motivate the rationale of focusing on tabular datasets with such sizes. Without detailed explantation, some claims in the paper might be confusing (h4aV: "while the authors claim that it is important to integrate LLM and GBDT as the dataset size increases, results such as Table 3 (Appendix) do not support this ...").

* Using Flan-T5 as the primary LLM for experiments "is not very convincing" (Fo2b). While the authors further provided results from Llama-8B, the worse results from this paper raises concerns on "once a better open LLM is released, LLM-Boost will not be able to benefit from it. (h4aV)". The authors shall experiment with more recently released LLMs with ~10B parameters in order to show the general applicability of LLM-Boost while studying the impact of different LLMs on the performance of the approach. Nit: such important results, including those from Llama-8B, shall be moved to the main text of the paper.

**Additional Comments On Reviewer Discussion:**

The two major issues as discussed in the meta review remain unsolved after the rebuttal. There are several other issues around clarity (Fo2b: "The major baseline is quite strange") and potential application with proprietary models like GPT-4 (Fo2b), which are addressed during the rebuttal phase.

---

### Decision · Program_Chairs · 2025-01-22

Reject